# UHRF1 is a mediator of KRAS driven oncogenesis in lung adenocarcinoma

Kaja Kostyrko [1] ✉, Marta Román[1], Alex G. Lee[1], David R. Simpson[1], Phuong T. Dinh[1], Stanley G. Leung [1], Kieren D. Marini[1], Marcus R. Kelly[2], Joshua Broyde[3], Andrea Califano[3,4,5,6,7], Peter K. Jackson [2] & E. Alejandro Sweet-Cordero [1] ✉

KRAS is a frequent driver in lung cancer. To identify KRAS-specific vulnerabilities in lung cancer, we performed RNAi screens in primary spheroids derived from a Kras mutant mouse lung cancer model and discovered an epigenetic regulator Ubiquitin-like containing PHD and RING finger domains 1 (UHRF1). In human lung cancer models UHRF1 knock-out selectively impaired growth and induced apoptosis only in KRAS mutant cells. Genome-wide methylation and gene expression analysis of UHRF1-depleted KRAS mutant cells revealed global DNA hypomethylation leading to upregulation of tumor suppressor genes (TSGs). A focused CRISPR/Cas9 screen validated several of these TSGs as mediators of UHRF1-driven tumorigenesis. In vivo, UHRF1 knock-out inhibited tumor growth of KRAS-driven mouse lung cancer models. Finally, in lung cancer patients high UHRF1 expression is anti-correlated with TSG expression and predicts worse outcomes for patients with KRAS mutant tumors. These results nominate UHRF1 as a KRAS-specific vulnerability and potential target for therapeutic intervention.

Lung cancer is the main cause of cancer-related death in the world and approximately 30% of non-small cell lung cancer (NSCLC) cases are driven by oncogenic KRAS[1]. The recent development of small molecules targeting specific KRAS alleles and the results of clinical trials in NSCLC patients using these agents have renewed interest in direct KRAS inhibition[2,3]. However, direct targeting of KRAS is currently feasible for only a subset of patients. Moreover, both intrinsic and acquired resistance to direct KRAS inhibition develops in most patients[4,5], indicating a continued need for complementary approaches to inhibit KRAS-driven oncogenesis. Prior efforts to target downstream KRAS effectors have shown limited efficacy in the clinic, due to incomplete inhibition, feedback pathway activation or toxicity[6]. An alternative avenue for targeting KRAS-driven cancer is to exploit oncogene-specific vulnerabilities. While genome-wide RNAi or

CRISPR/Cas9 screens have been widely applied in the search for KRAS-specific dependencies[7–9], these screens have mostly been limited to well-established cell lines cultured in standard two-dimensional (2D) setting, potentially missing vulnerabilities that arise in the context of other growth conditions. In contrast, three-dimensional (3D) cultures recapitulate more features of in vivo tumors and thus may be a better model for studying cancer biology[10,11]. Moreover, 3D cultures have been found to be enriched in tumor-propagating cells (TPCs), a subset of cells driving tumor initiation, maintenance and progression[12–14]. Despite these observations, limited efforts have been made to identify KRAS vulnerabilities using primary cells or cells grown in 3D culture.

Although functional genomic screens using primary tumor cells in 3D are technically challenging and have limited throughput, we reasoned that such a screening strategy could have a unique potential to

[1]Division of Oncology, Department of Pediatrics, University of California, San Francisco, San Francisco, CA, USA. [2]Baxter Laboratory, Department of Microbiology and Immunology, Stanford University School of Medicine, Stanford, CA, USA. [3]Department of Systems Biology, Columbia University, New York, NY, USA. [4]Department of Biomedical Informatics, Columbia University, New York, NY, USA. [5]Herbert Irving Comprehensive Cancer Center, Columbia University, New York, NY, USA. [6]Department of Biochemistry and Molecular Biophysics, Columbia University, New York, NY, USA. [7]Department of Medicine, Vagelos College of Physicians and Surgeons, Columbia University, New York, NY, USA. ✉e-mail: kaja.kostyrko@ucsf.edu; alejandro.sweet-cordero@ucsf.edu

discover novel dependencies relevant for cancers driven by oncogenic KRAS that have not been observed in 2D screens. Therefore, we performed RNAi screens in 3D cultures of primary cells derived from a genetically engineered mouse model (GEMM) of lung cancer driven by the activation of Kras and loss of p53[15,16]. We used pooled shRNA libraries targeting 115 murine homologs of known and predicted KRAS interactors and effector genes[17,18]. In parallel, we screened Kras mutant murine lung cancer cells grown as monolayers to determine 3D-specific vulnerabilities. These screens identified *Ubiquitin-like with PHD and ring finger domains 1* (*Uhrf1*) as a gene uniquely essential for the growth of primary lung cancer spheroids.

UHRF1 is E3 ubiquitin ligase highly expressed in many human cancers compared to normal tissues, and has been linked to rapid disease progression[19–25]. UHRF1 is best characterized for its role in DNMT1-mediated methylation of hemi-methylated DNA in the S phase of the cell cycle[26–29]. UHRF1 has also been suggested to be necessary for the maintenance of DNA methylation throughout the cell cycle in colorectal cancer[21,30]. UHRF1 likely also has other effects as it has been shown to promote DNA double-strand break repair through a direct interaction with BRCA1[31] and by mediating recruitment of DNA repair factors in the Fanconi Anemia pathway to interstrand crosslinks (ICLs)[32].

Here, we show that UHRF1 loss in KRAS mutant cells leads to widespread DNA hypomethylation and overexpression of tumor suppressor genes (TSGs). Using a focused CRISPR/Cas9 screen, we identify candidate TSGs whose upregulation is most likely involved in the phenotypic response to UHRF1 loss in cells expressing oncogenic KRAS. In GEMM and xenograft models of KRAS-driven lung cancer, homozygous UHRF1 loss significantly decreases tumor growth and extends survival. Lastly, analysis of patient datasets demonstrates that high UHRF1 expression in patients with tumors harboring KRAS mutations is associated with poor prognosis. These results demonstrate that UHRF1 plays a critical role in KRAS-driven tumorigenesis and may be an attractive drug target for the treatment of KRAS mutant NSCLC and/or other KRAS-driven cancers.

## Results
### Loss-of-function screens identify Uhrf1 as essential for the growth of primary mouse lung cancer spheroids
To identify KRAS-specific vulnerabilities in lung cancer, we performed functional genomic screens in primary tumor cells derived from a

genetically engineered mouse model of lung cancer driven by the activation of the Kras G12D allele and the loss of p53 (Kras^LSL-G12D, Trp53^fl/fl; KP)[15,16] (Fig. 1a). We used pooled shRNA libraries containing 518 shRNAs targeting 115 genes (2–5 shRNAs per gene). Targets included downstream effectors and upstream modulators of KRAS inferred by the algorithms ARACNe and VIPER[33,34] and KRAS interactors identified using affinity purification mass spectrometry (AP/MS)[17] (Supplementary Data 1–3). Tumor cells were isolated from mouse lungs, infected with lentiviral libraries, and plated into matrigel-based 3D culture (see Methods). To identify genes specifically required for KRAS-driven oncogenesis in primary cancer spheroids, we performed a parallel screen using a mouse lung cancer cell line (LKR10), previously derived from the Kras G12D mouse model, cultured in standard 2D conditions[35].

Among the targeted genes, the knock-down of *Ubiquitin-like with PHD and ring finger domains 1* (*Uhrf1*) had a very strong effect in primary spheroids, with 4/4 shRNAs having a deleterious phenotype (Fig. 1b, Supplementary Fig. 1a, and Supplementary Data 4). In contrast, Uhrf1 knock-down had almost no effect in adherent LKR10 cells. In line with these observations, UHRF1 did not score in the DepMap or Project DRIVE RNAi screens[8,9] performed across panels of human lung cancer cell lines cultured in 2D (Supplementary Fig. 1b). Of note, UHRF1 was not included in the CRISPR DepMap libraries nor in the genome-wide CRISPR library used in our previous work[10,17], therefore we were unable to establish if complete knock-out of *UHRF1* in 2D could be a dependency in KRAS mutant lung cancer cell lines.

### UHRF1 loss impairs 3D growth of KRAS mutant human lung cancer cell lines
To extend these observations and determine if UHRF1 loss affects 3D growth of human NSCLC cells, we used CRISPR/Cas9 to knock-out *UHRF1* using two sgRNAs in a panel of human lung cancer cell lines (Fig. 2a and Supplementary Fig. 2a). To evaluate the role of UHRF1 in non-tumorigenic human cells, we also deleted *UHRF1* in immortalized human bronchial epithelial cells (HBECs). Loss of UHRF1 strongly impaired 3D growth of NSCLC cells expressing oncogenic KRAS while having little effect in KRAS wild-type cells (H1437) or in the non-transformed HBEC line (NL20 cells) (Fig. 2b). Both sphere number and sphere size were significantly decreased after *UHRF1* knock-out in cell lines expressing oncogenic KRAS (Fig. 2c). We previously established

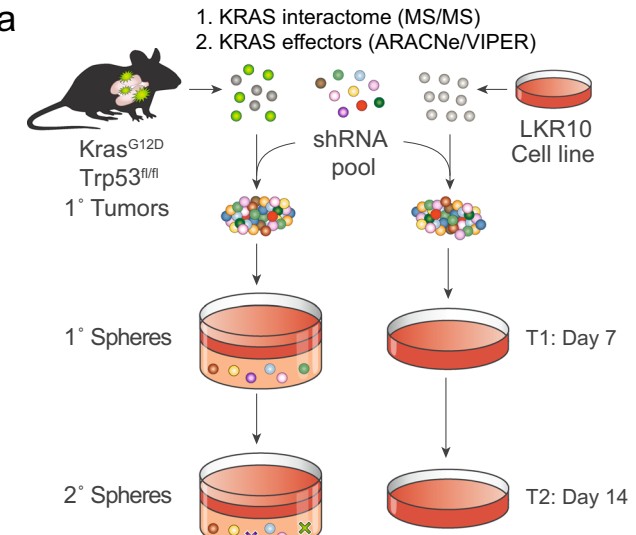

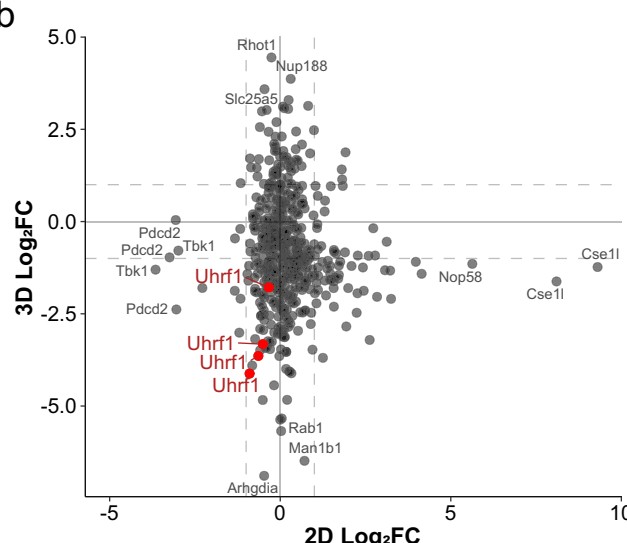

**Fig. 1 | Loss of Uhrf1 inhibits 3D growth of primary mouse NSCLC spheroids.**
**a** Schematic representation of the pooled RNAi screens in primary spheroids and LKR10 cells. **b** Plot of shRNAs depleted from the cell population between T2 and T1 in primary mouse spheroids (y-axis) and murine LKR10 cells grown as monolayers (x-axis). Change in shRNA representation is shown as log2 fold change (log2FC). Uhrf1-specific shRNAs are indicated in red.

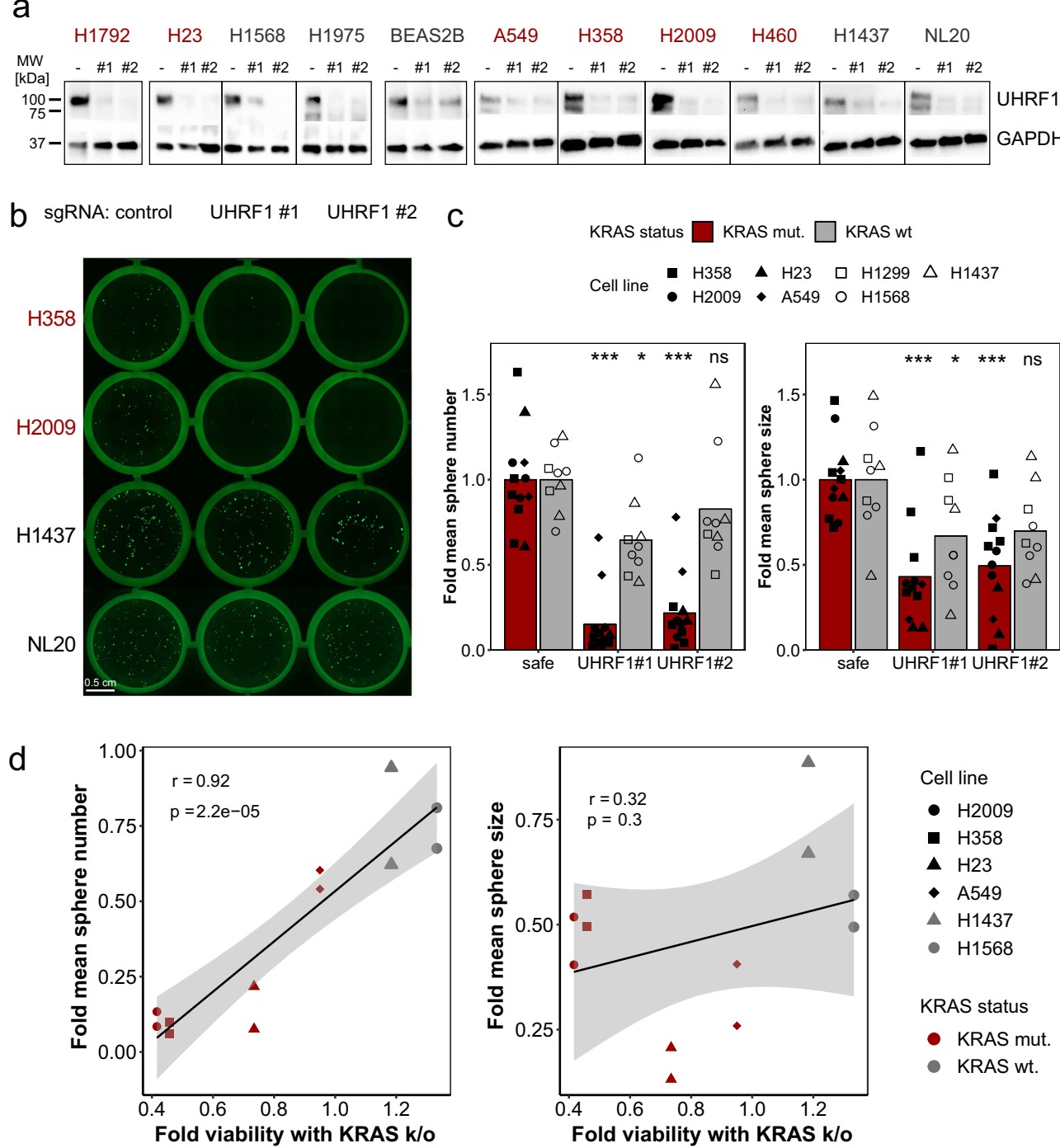

**Fig. 2 | CRISPR/Cas9 knock-out of UHRF1 inhibits 3D growth of KRAS-dependent human lung cancer cells. a** Representative western blot images showing loss of the UHRF1 protein with CRISPR/Cas9 knock-out of the UHRF1 gene. Safe-cutting control sgRNA indicated with -, two sgRNAs against UHRF1 indicated with #1 and #2. Cell lines with a KRAS mutation indicated in red, KRAS wild-type cell lines in gray font. The experiment was repeated six times. **b** Representative images of GFP-expressing spheroids from three NSCLC cell lines (H2009, H358, H1437) and one HBEC cell line (NL20) expressing Cas9 and the indicated sgRNAs. Control - safe-cutting control sgRNA, UHRF1 #1 – UHRF1 sgRNA #1, and UHRF #2 – UHRF1 sgRNA #2. The experiment was repeated seven times. **c** Quantification of sphere number (left) and size (right) in KRAS mutant (H23, H358, H2009, A549) and KRAS wild-type (H1568, H1437, H1299) cell lines. Bars represent means, points represent $n = 12$ (KRAS mutant cells) or $n = 8$ (KRAS wild-type cells) individual biological replicates; *$p < 0.05$, ***$p < 0.001$, ns - not significant by anova followed by Dunnett's multiple comparisons test between the indicated UHRF1 sgRNA and the control sgRNA. **d** Correlation between sphere numbers (left) and size (right) with UHRF1 knock-out and viability with KRAS knock-out. Correlation coefficient and *p*-value computed using Pearson's product-moment correlation test. Linear trend lines were generated using a linear model, shaded confidence regions represent CI = 0.95. Source data are provided as a Source data file.

the degree of KRAS-dependence in these cell lines using CRISPR/Cas9 knock-out[17]. Sphere number after *UHRF1* knock-out was positively correlated with the degree of KRAS dependence ($R = 0.92$, $p < 0.001$) (Fig. 2d), while the correlation between KRAS dependence and sphere size with UHRF1 loss was not statistically significant ($R = 0.32$, $p = 0.3$). This may indicate that UHRF1 is essential for the initial phases of KRAS-driven tumor growth, while the maintenance of spheroid growth may be less directly dependent on UHRF1 activity.

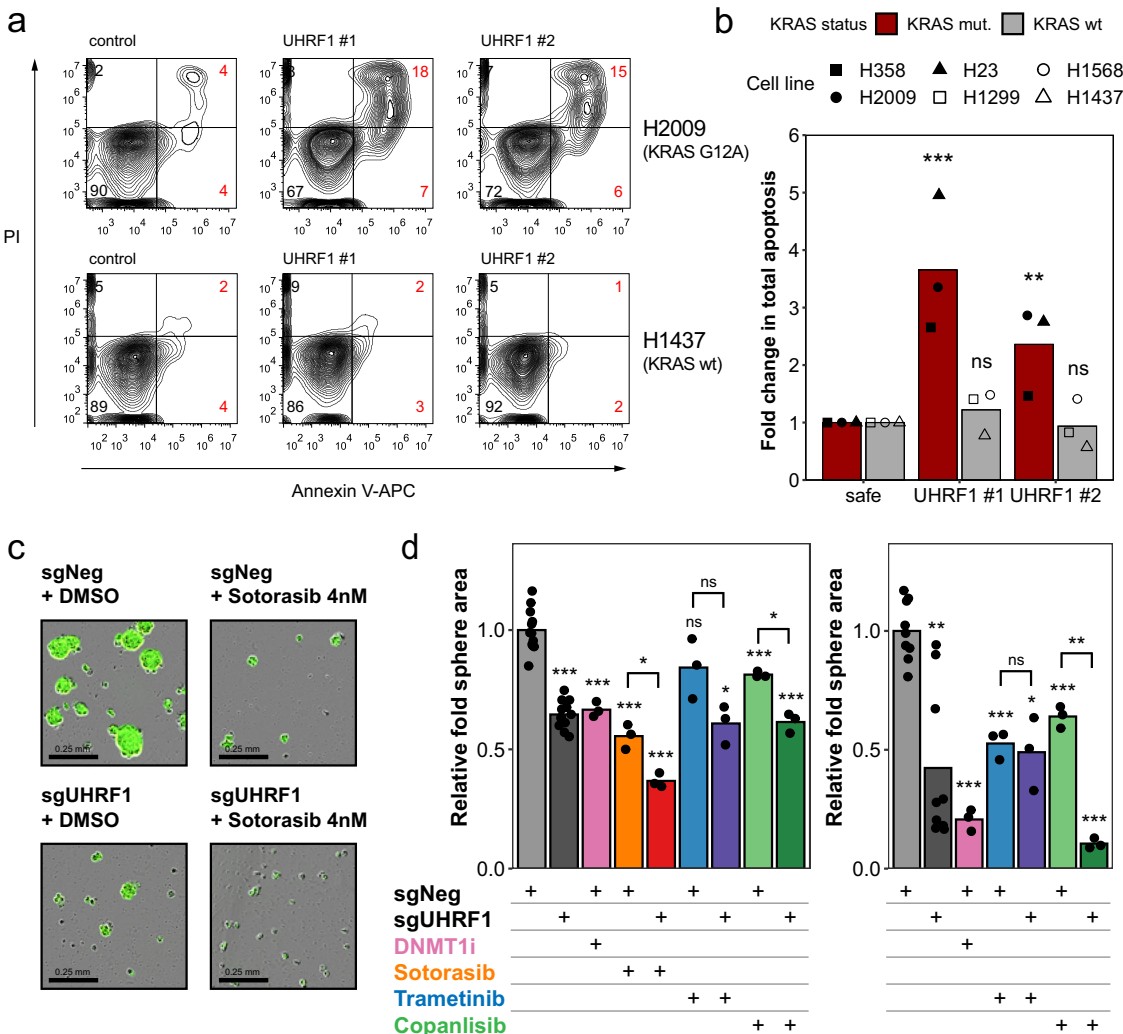

**Fig. 3 | Loss of UHRF1 leads to apoptosis in cells expressing oncogenic KRAS cells. a** Flow cytometry of KRAS mutant (H2009) and KRAS wild-type (H1437) cells expressing Cas9 and transduced with lentiviral vectors expressing the indicated sgRNAs. Cells were co-stained with Annexin V-APC and propidium iodide (PI). **b** Quantification of apoptosis in a panel of lung cancer cell lines. Red bars – KRAS mutant cells (H2009, H358, H23); gray bars – KRAS wt cells (H1568, H1437, H1299). Bars represent means of $n = 3$ of biological replicates; **$p = 0.007954$, ***$p = 0.000206$, not significant (ns) $p > 0.05$ by anova followed by Dunnett's multiple comparisons test between the indicated UHRF1 sgRNA and the control sgRNA. **c** Images of spheroids from H358 cells expressing Cas9 and the indicated sgRNAs

treated with either DMSO or 4 nM of Sotorasib. **d** Quantification of spheroid area in two cell lines (left – H358, right – A549) expressing Cas9 and the indicated sgRNAs. Cells were treated with DMSO, 10 nM of DNMT1i GSK3685032, 4 nM of Sotorasib, 3 nM of MEKi Trametinib, or 15 nM of PI3Ki Copanlisib. Bars represent mean of $n = 4$ (H358 sgNeg+DMSO, H358 sgUHRF1+DMSO), $n = 3$ (A549 sgNeg+DMSO, A549 sgUHRF1+DMSO), or $n = 1$ biological replicates (all remaining conditions) performed in $n = 3$ technical replicates. Bars represent means and are presented as fold change relative to control (sgNeg); *$p < 0.05$, **$p < 0.01$, ***$p < 0.001$, ns - not significant by unpaired two-sided Student's $t$ test. Source data are provided as a Source data file.

In contrast to the knock-down experiments, CRISPR-driven *UHRF1* knock-out also significantly impaired 2D proliferation and colony formation of human NSCLC cell lines carrying oncogenic KRAS (Supplementary Fig. 2b–d). These results suggest that incomplete loss of UHRF1 is particularly deleterious to cells in 3D, whereas complete loss is deleterious to cells in both 2D and 3D. Importantly, UHRF1 loss had a weaker effect in KRAS wild-type cells and HBECs. Quantification of apoptosis by flow cytometry demonstrated that the loss of UHRF1 induces cell death in cells expressing oncogenic KRAS, while having no effect in KRAS-independent cells or HBECs (Fig. 3a, b and Supplementary Fig. 2e).

**UHRF1 loss synergizes with pharmacological inhibition of KRAS and its downstream effectors**

Given the above results, we explored if UHRF1 depletion would sensitize cells to direct inhibition of KRAS or its downstream effectors

MEK or PI3K. We treated two of the KRAS mutant cell lines (H358, A549) with the MEK inhibitor trametinib or the PI3K inhibitor copanlisib and measured 3D growth of these cells over time. In addition, H358 cells were treated with the KRAS G12C inhibitor sotorasib. Treatment of control H358 cells with sotorasib, and to a lesser extent with trametinib and copanlisib, decreased spheroid growth comparably to the genetic inhibition of UHRF1 (Fig. 3c, d and Supplementary Fig. 2f). We also observed the same effect with trametinib and copanlisib in A549 cells (Fig. 3d). When combined with *UHRF1* knock-out, sotorasib and copanlisib treatment resulted in a significantly stronger growth inhibition, suggesting that UHRF1 depletion can synergize with direct KRAS or PI3K inhibition. Minimal combinatorial effect of UHRF1 depletion was observed with trametinib treatment in H358 and A549 cells.

As UHRF1 is known to play a role in DNMT1-mediated methylation, we hypothesized that DNMT1 inhibition might phenocopy the loss of

UHRF1. Consistent with this hypothesis, the treatment of the two KRAS mutant cell lines with the DNMT1-selective inhibitor GSK3685032[36] resulted in a significant decrease in spheroid growth, which was similar to that of *UHRF1* knock-out (Fig. 3d). This suggests that the observed anti-tumor effect of UHRF1 depletion is, at least partially, mediated by its role in DNA methylation, and that DNMT1 inhibition may mimic the effect of UHRF1 loss.

## UHRF1 loss leads to hypomethylation and overexpression of lung cancer-specific tumor suppressor genes

UHRF1 is highly expressed in many cancers, including lung adeno-carcinoma, compared to normal tissues[19]. Analysis of UHRF1 RNA and protein expression in human NSCLC cell lines and patient lung adeno-carcinoma samples from TCGA demonstrated a tendency towards higher UHRF1 expression in KRAS mutant cell lines (Supplementary Fig. 3a, b). To test if UHRF1 could be a direct effector of oncogenic KRAS in lung cancer, we depleted KRAS in lung cancer cell lines and quantified the levels of UHRF1 protein. Upon *KRAS* knock-down the levels of UHRF1 protein decreased primarily in KRAS mutant cell lines (Supplementary Fig. 3c). However, we also observed a concomitant decrease in Cyclin B1, a marker of G2 and M cell cycle phases, and Cyclin D1, which is required for the G1/S transition, consistent with KRAS loss leading to cell cycle arrest in these cells (Supplementary Fig. 3c, d). Flow cytometry analysis confirmed a G1-phase arrest in these cell lines (Supplementary Fig. 3e). In addition, we also assessed if the levels of UHRF1 protein would be affected by inhibition of KRAS, or its downstream effectors MEK and PI3K. We performed immunoblotting with an anti-UHRF1 antibody on protein extracts from H358 cells treated with sotorasib, trametinib or copanlisib. In these cells UHRF1 protein levels were significantly affected by sotorasib and trametinib and, to a lesser extent copanlisib (Supplementary Fig. 3f). Also in this case, in addition to decreased UHRF1 protein levels, we observed a concomitant decrease in Cyclin D1, pointing to a G1 cell cycle arrest. Given that UHRF1 expression has been shown to be highest in S and G2 phases[31], the observed decrease in UHRF1 protein may be a consequence of cell cycle arrest in G1 rather than a direct result of KRAS depletion or inhibition.

We also noted that in KRAS mutant cells UHRF1 knockdown resulted in an expansion of the S phase of the cell cycle. Given that UHRF1 was previously shown to regulate DNA double-strand break repair pathway choice in S phase[31], we hypothesized that the loss of UHRF1 in these cells may lead to an accumulation of un-resolved DNA double-strand breaks. Consistent with this, we observed an increased level of phosphorylated histone 2AX (pH2AX), a marker for DNA damage, in UHRF1-depleted cells compared to control cells (Supplementary Fig. 3g–j).

To assess the impact of UHRF1 loss on DNA methylation in KRAS mutant lung cancer cells, we depleted UHRF1 or KRAS using siRNA in two KRAS-driven cell lines (H358 and A549) and analyzed changes in DNA methylation. We also collected RNA from matched samples for gene expression analysis. Loss of UHRF1 strongly affected global DNA methylation in both cell lines compared to control (Fig. 4a and Supplementary Fig. 4a, b). Gene ontology and pathway analysis on differentially methylated regions revealed that the regions hypomethylated with UHRF1 loss were strongly enriched for genes involved in the regulation of small GTPase activity, transcription, cell cycle and development (Fig. 4b, Supplementary Fig. 4c, and Supplementary Data 5). RNA sequencing analysis of matched samples revealed significant gene expression changes in both UHRF1 knock-down and KRAS knock-down cells compared to the control cells (Fig. 4c and Supplementary Fig. 4d). Differential gene expression analysis identified 2892 significantly upregulated and 2759 significantly downregulated genes with UHRF1 loss (Fig. 4d and Supplementary Data 6). KRAS depletion resulted in more pronounced gene expression changes with 4971 upregulated and 4174 downregulated genes (Supplementary Fig. 4e and Supplementary Data 7). A significant

proportion of genes differentially expressed upon UHRF1 knock-down were also differentially expressed after KRAS knock-down ($p < 0.00001$ for both up- and downregulated gene subsets; Fig. 4e).

We subsequently performed gene set enrichment analysis (GSEA) on genes differentially expressed in UHRF1 knock-down cells (see Methods) and visualized the most significantly up- and downregulated pathways (FDR < 0.05) using Enrichment Map[37] (Supplementary Fig. 4f). Some of the most downregulated terms included cell cycle checkpoints, DNA-damage response and ribosome biogenesis, likely a consequence of cell cycle arrest and apoptosis program activated by UHRF1 loss. The most significantly upregulated were genes involved in transmembrane transport of ions and organic acids, ion homeostasis, and Golgi and ER-related pathways, suggesting that metabolic reprograming may be a response to UHRF1 depletion. Of note, receptor tyrosine kinase (RTK) signaling, including signaling via EGFR, was also upregulated in response to UHRF1 loss.

We next inferred activity of known cancer-related pathways from gene expression data using PROGENy[38]. In both cell lines, UHRF1 knock-down lead to an increase in the activity of the TRAIL pathway, a potent stimulator of apoptosis, as well as increase in the hypoxic response and PI3K signaling pathways (Supplementary Fig. 4g). In A549 cells loss of both UHRF1 and KRAS also resulted in decreased activity of the WNT and p53 pathways. As expected, KRAS knock-down led to a significant decrease in the activity of the EGFR and MAP kinase signaling. The activity of these pathways was not strongly affected by UHRF1 knock-down in either cell line, suggesting that they are not regulated by UHRF1.

To determine whether UHRF1 regulates tumor suppressor gene (TSG) expression in KRAS-driven lung cancer we performed GSEA using a gene set of TSGs known to be down-regulated in lung cancer (https://bioinfo.uth.edu/TSGene/). This lung-cancer-specific TSG set was significantly enriched in UHRF1-depleted samples (Supplementary Fig. 4h). Comparison between TSGs significantly upregulated after UHRF1 loss and TSGs significantly hypomethylated in UHRF1-depleted cells identified 80 genes in common (Fig. 4f and Supplementary Data 8) including several tumor suppressor genes frequently mutated in human cancers (*PTEN, DLC1, CUX1, DUSP22*), negative regulators of pro-tumorigenic WNT/β-catenin pathway (*FHL1, MCC, AXIN2, WNT7A, CDH13, RASSF8*), apoptosis-activators (*HIPK2, CD82, GAS1*), and negative regulators of mTORC1 signaling (*TSC1, TMEM127*). Many of these TSGs were also overexpressed with KRAS knock-down (Supplementary Fig. 4e, i). Examples of CpG methylation in the promoter regions of four of these TSGs (*HIPK2, PTPN1, WFDC1, MFSD2A*) are shown in Supplementary Fig. 4j.

To identify which of the 80 TSGs mediate the UHRF1 phenotype in KRAS mutant cancer cells, we designed a focused CRISPR library targeting those genes and used this library to perform a screen in Cas9-expressing A549 cells treated with either a non-targeting control sgRNA or an *UHRF1*-targeting sgRNA (Fig. 5a, Supplementary Fig. 5a, b, and Supplementary Data 9). We reasoned that sgRNAs targeting TSGs regulated by UHRF1 would rescue the apoptotic phenotype driven by UHRF1 loss in KRAS mutant cells leading to positive selection of these sgRNAs in UHRF1-depleted cells but not in control cells. We found that the majority of the TSGs included in the library (64/80) rescued proliferation defects seen in *UHRF1* knock-out cells (z-score > 0), while in control cells knock-out of most of the genes had no or negative effect on proliferation (54/80, z-score ≤ 0; Fig. 5b, c and Supplementary Data 10). Among the 64 genes that positively scored in UHRF1-depleted cells, we found 15 genes (*EMP2, CSRNP1, ZDHHC2, PAFAH1B1, CD44, GABARAP, EPHA3, NR4A1, SRGAP3, CHST10, PIN1, HIPK2, ZNF185, TMEM127, CBL*), whose knock-out significantly improved proliferation only in UHRF1 knock-out cells, while having little effect in control cells (Fig. 5d), suggesting that these TSGs may be regulated by UHRF1. In line with this, RNA sequencing revealed that expression of these 15 genes was increased with UHRF1 knock-down in both A549 and H358

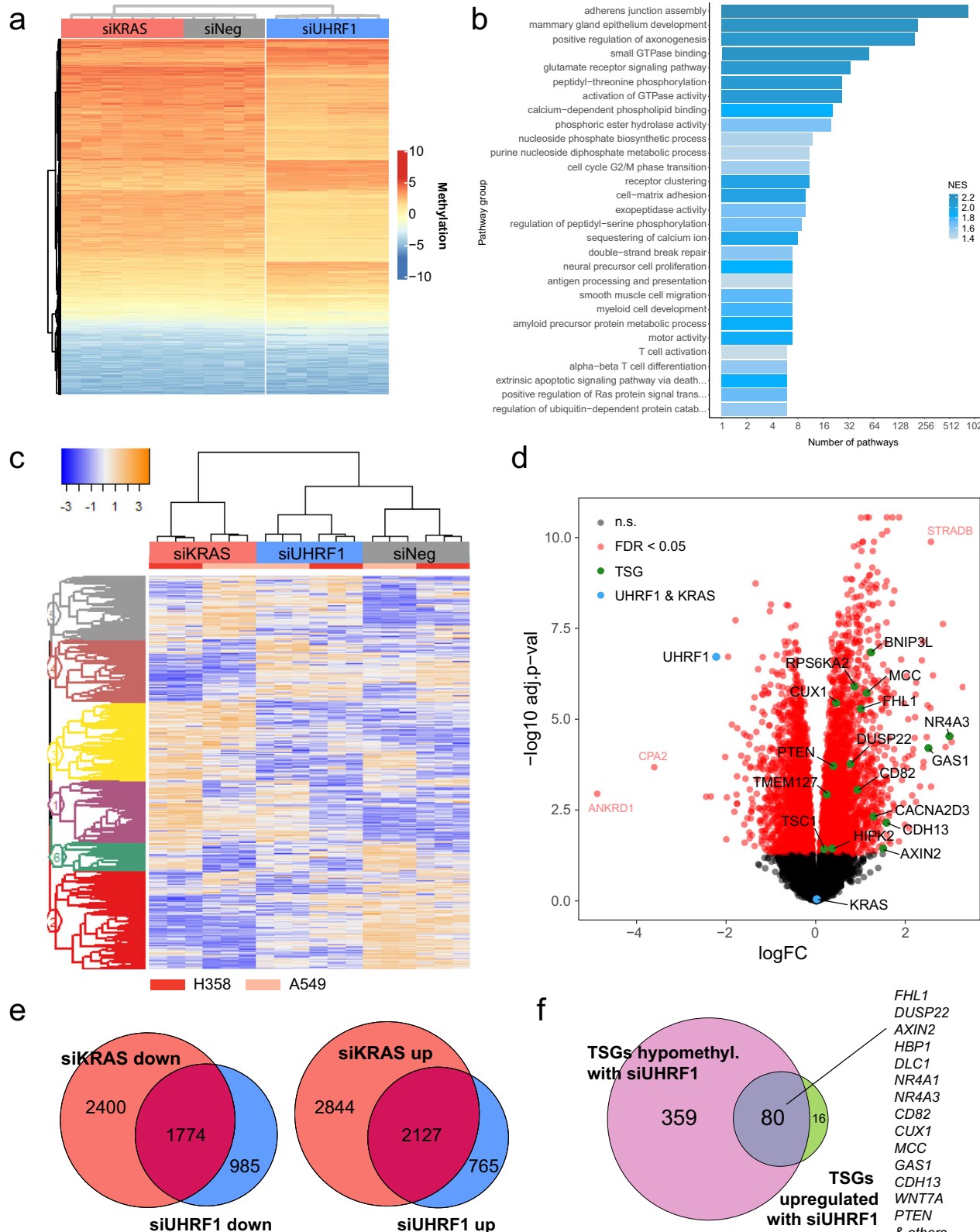

**Fig. 4 | UHRF1 loss leads to re-expression of tumor suppressor genes.**
**a** Hierarchical clustering of differentially methylated regions in two KRAS mutant NSCLC cell lines (H358, A549) transfected with siRNAs against UHRF1, KRAS, or a negative control siRNA (siNeg). **b** Gene set enrichment analysis (GSEA) using GO terms, KEGG and Reactome gene sets on CpGs differentially methylated between cells treated with siUHRF1 compared to control cells. Plots show the top aggregated pathways enriched in hypomethylated genes ("down") ranked by gene ratio (gene/total gene) for a given pathway. **c** Hierarchical clustering of differential gene expression in two KRAS mutant NSCLC cell lines (H358, A549) transfected with

siRNAs against UHRF1, KRAS, or siNeg. **d** Volcano plot of differential gene expression between siUHRF1 and siNeg in two KRAS mutant NSCLC cell lines (H358, A549). Green points – examples of significantly overexpressed tumor suppressor genes (TSGs), blue point – UHRF1 and KRAS. **e** Venn diagrams of significantly (FDR < 0.05) downregulated (left) or upregulated (right) genes in cell treated with siUHRF1 or siKRAS. Both up- and downregulated genes show significant overlap ($p < 0.00001$, hypergeometric test) between siUHRF1 and siKRAS. **f** Venn diagram of tumor suppressor genes (TSGs) hypomethylated with siUHRF1 (EPIC methylation array dataset) and TSGs upregulated with siUHRF1 (RNAseq dataset) in A549 and H358 cells.

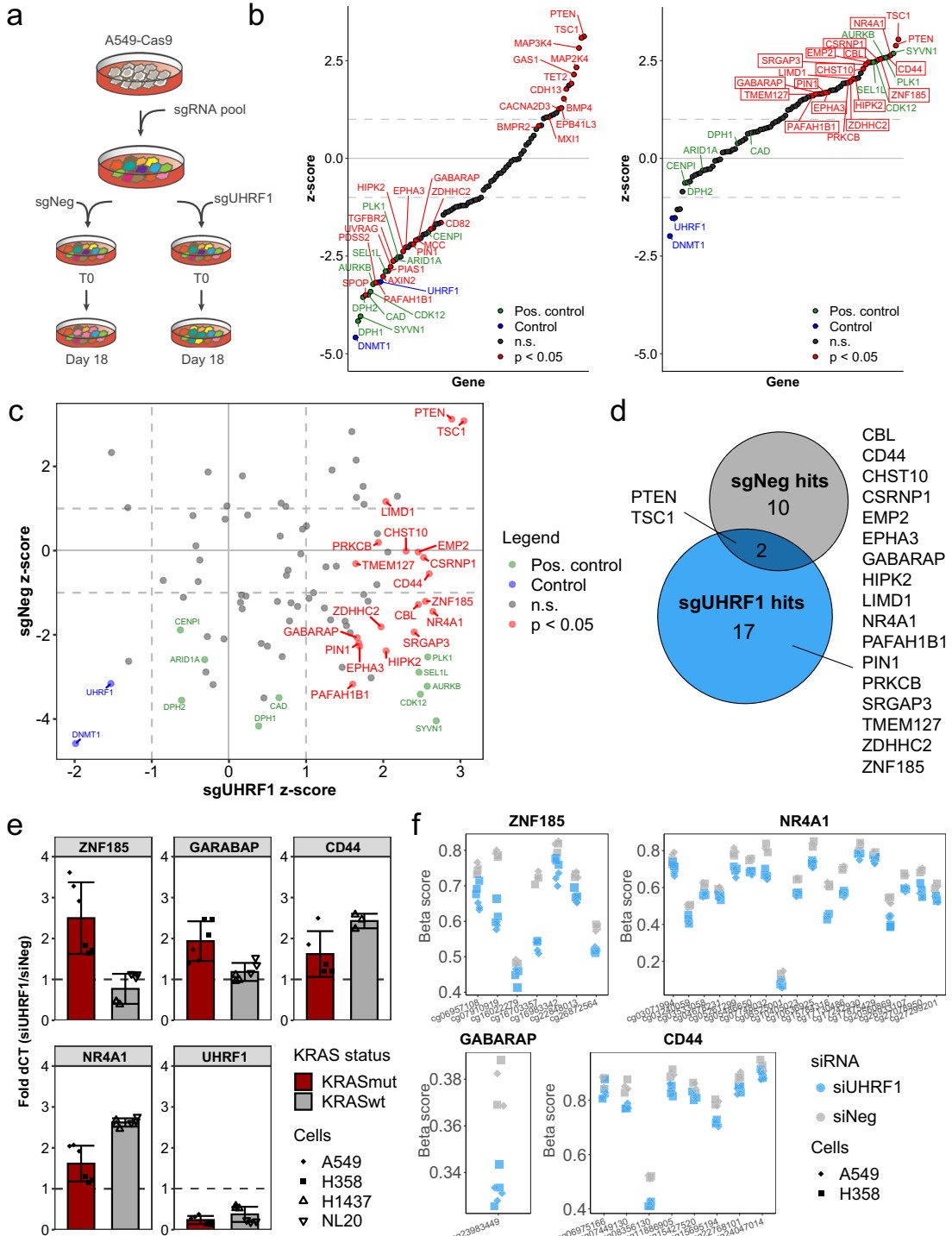

**Fig. 5 | CRISPR screen for UHRF1-dependent tumor suppressor genes. a** Screen setup in A549 Cas9-expressing cells. **b** Screen results in control (sgNeg, left) cells and UHRF knock-out cells (sgUHRF1, right). Genes ranked by increasing *z*-score. Genes labeled in green – positive control, blue – UHRF1 and DNMT1, red – statistically significant hits (*p*-value < 0.05). In red frames – genes with positive effect on growth in UHRF1 knock-out cells but with negative or no effect in control (sgNeg) cells. **c** Scatterplot of *z*-scores in control (sgNeg) samples versus *z*-scores in UHRF1 knock-out (sgUHRF1) samples. Green – positive control genes, blue – UHRF1 and DNMT1, red – genes with *p*-value < 0.05 in UHRF1 knock-out cells. **d** Venn diagram of positive hits identified in sgUHRF1 and in sgNeg conditions (excluding control genes). Only hits with *p*-value < 0.05 were included. Hits in UHRF1-depleted cells are listed. **e** RNA expression of four UHRF1-specific tumor suppressor genes identified in the minipool CRISPR screen in two KRAS mutant cell lines (red bars – H358, A549)

and two KRAS wild-type cell lines (gray bars – H1437, NL20) treated with control (siNeg) or UHRF1 siRNA (siUHRF1). Expression of UHRF1 is shown as control of knock-down efficiency. Bars represent mean fold expression of the indicated gene in siUHRF1-treated cells relative to cells treated with a negative control siRNA (siNeg); error bars represent standard deviation; points represent technical replicates (*n* = 3) from two biological replicates. **f** Plots show probes in the EPIC methylation array with statistically significant hypomethylation (adjusted *p*-value < 0.05) in regions related to the promoter (annotated as promoter-associated, 5' UTR, 3' UTR, TSS, and/or related to the 1st exon of the gene). The y-axis represents beta values color-coded by siRNA treatment (blue – siUHRF1, gray – siNeg). Shape of the data points represents cell line. Source data are provided as a Source data file.

cells (Supplementary Fig. 5c). In addition, we also used real-time PCR to analyze the expression of four of these genes in H358, A549, H1437, and NL20 cell lines (Fig. 5e), and found that two of these genes (ZNF185, GARABAP) were regulated by UHRF1 only in KRAS mutant cells, while the other two (CD44, NR4A1) appeared to be more universal UHRF1 targets. Finally, methylation analysis of the 15 UHRF1-specific hits identified in the CRISPR screen also revealed significant hypomethylation in the promoter and regulatory regions of these genes in UHRF1-depleted cells (Fig. 5f and Supplementary Fig. 5c), suggesting that their expression may be the regulated by UHRF1-mediated methylation.

## Loss of Uhrf1 in KRAS-driven models of lung cancer inhibits in vivo tumor growth

To test the consequence of Uhrf1 loss on tumor formation in vivo, Kras$^{LSL-G12D/+}$ Trp53$^{fl/fl}$ (KP) mice were crossed with mice harboring loxP sites flanking exon 4 of Uhrf1 (Uhrf1$^{fl/fl}$ mice)[39] to generate UKP mice. Uhrf1$^{fl/fl}$ mice were also crossed to Kras$^{LSL-G12D/+}$ Trp53$^{+/+}$ (K) mice to obtain UK mice. To initiate tumor formation, Cre-expressing adenovirus was delivered intranasally at 4–10 weeks of age. At 12–16 weeks after tumor initiation, mice were euthanized and lungs were assessed for tumor burden. Homozygous deletion of Uhrf1 with concomitant activation of the Kras G12D allele and loss of Trp53 led to a significant decrease in tumor formation compared to Uhrf1 wild-type mice (Fig. 6a, c). The same was true when comparing K and UK mice, a less aggressive lung adenoma model driven by Kras G12D alone (Fig. 6b, c). In both UK and UKP models, loss of Uhrf1 resulted in fewer and smaller lung lesions. Homozygous Uhrf1 knock-out also significantly extended the survival of UKP mice compared to Uhrf1 heterozygous or wild-type KP mice (Fig. 6d). Uhrf1 immunostaining of tumor-bearing lungs of KP and UKP mice revealed the presence of Uhrf1 protein in all cancer lesions (Fig. 6e and Supplementary Fig. 6a), suggesting incomplete Uhrf1 gene inactivation by the Cre recombinase in the UKP mice. Quantification of Uhrf1-specific fluorescence signal intensity in immunofluorescence images from KP and UKP mice suggested that a lower amount of Uhrf1 protein was present in UKP lesions compared to KP (Supplementary Fig. 6b), further supporting this hypothesis.

Uhrf1 was previously shown to be expressed only in dividing cells in normal mouse lungs[40]. To determine whether this was also true in lung cancer cells, tissue sections from either KP mouse tumor-bearing lungs or human lung patient derived xenograft (PDX) model driven by the KRAS G12C allele were co-stained with antibodies against UHRF1 and Ki67. Normal testicular mouse tissue was used as a control. In normal mouse testis a near complete co-localization of Uhrf1 and Ki67 was observed, while in the two cancer samples (KP mouse and human PDX) UHRF1 was also present in Ki67-negative non-dividing cells (Supplementary Fig. 6c-d). Similar results were obtained by Uhrf1 and EdU co-immunostaining of lungs from KP mice treated with EdU (Fig. 6f and Supplementary Fig. 6e). Moreover, in these mice we observed a decreased proportion of Uhrf1/EdU double-positive cells and a concomitant increase in Uhrf1 only positive cells compared to UKP mouse lesions that escaped Uhrf1 deletion or to normal mouse lungs (Fig. 6g and Supplementary Fig. 6f). These results suggest a deregulation of Uhrf1 expression in lung cancer.

Finally, we derived a cell line from tumor-bearing lungs of a UKP mouse. This UKP cell line retained expression of Uhrf1 protein (Fig. 6h), again suggesting that tumor development in the UKP model is due to incomplete Cre-mediated deletion of UHRF1. In vitro treatment of these cells with Cre-expressing adenovirus resulted in complete loss of Uhrf1 protein within 4 days of treatment. In line with observations from human KRAS mutant cell lines, Cre-mediated in vitro knock-out of Uhrf1 in UKP cells decreased colony growth in 2D and 3D compared to the empty adenovirus control (Supplementary Fig. 6g), further supporting a critical role for UHRF1 in KRAS-driven lung cancer.

We next assessed the impact of UHRF1 loss on the in vivo growth of human KRAS-driven lung adenocarcinoma cells using a competitive growth assay in a xenograft model of lung cancer. A549 cells expressing Cas9, GFP and an sgRNA targeting UHRF1 or a control sgRNA were mixed 1:1 with A549 cells carrying a control sgRNA vector labeled with mCherry and subsequently injected subcutaneously into mice. Flow cytometry measurement of GFP:mCherry ratios in samples prior to implantation and at the end of study revealed that the knock-out of UHRF1 with two different sgRNAs leads to a significant decrease in tumor growth (Fig. 7a, b). In line with the in vitro data (Figs. 2 and 3 and Supplementary Fig. 2), the more effective sgRNA #1 led to a complete inhibition of tumor growth, while sgRNA #2 had a weaker effect.

## High UHRF1 expression is a marker of poor prognosis in human KRAS mutant LUAD

Given the in vitro and in vivo data pointing to a role of UHRF1 in KRAS-driven lung cancer, we hypothesized that human lung cancer patients with elevated UHRF1 levels may have a worse prognosis than patients with low UHRF1 expression. Thus, we performed survival analysis using the lung adenocarcinoma (LUAD) tumor dataset from the cancer genome atlas (TCGA) stratified by UHRF1 expression (high vs normal; Fig. 8a). Higher UHRF1 expression was associated with worse disease specific survival (DSS) in patients with tumors carrying oncogenic KRAS (HR: 2.85, p = 0.01), while the association was much weaker and not statistically significant in the KRAS wild-type cohort (HR: 1.53, p = 0.092). These results further support an important role for UHRF1 in LUAD and a particular dependance of KRAS mutant tumors on UHRF1 expression.

We also assessed the relationship between UHRF1 expression and the expression of previously published lung cancer-specific tumor suppressor genes[41] and those from the TSGene database in lung cancer patient samples from TCGA. Expression of over 84% (450/537) of these TSGs was significantly anticorrelated with UHRF1 expression (Fig. 8b, Supplementary Fig. 7a, and Supplementary Data 11), including many of the TSGs identified as top hits in the CRISPR minipool screen in UHRF1 knock-out cells, such as EMP2, CSRNP1, NR4A1, or CD44 (Fig. 8c). This suggests that high expression of UHRF1 may drive TSG silencing and thus contribute to lung cancer progression. Moreover, out of 450 TSGs that were significantly anticorrelated with UHFR1, 323 were also significantly correlated with KRAS expression, of which 90% (291/323) was anticorrelated. Thus, we hypothesize that therapeutic targeting UHRF1 could lead to reactivation of these TSGs and effectively restrain tumor development.

Finally, from the subset of genes that were anticorrelated with both KRAS and UHRF1 expression we selected genes whose increased expression was protective (hazard ratio <1) in KRAS mutant lung cancer patients. This led to the identification of a set of 16 TSGs whose high expression correlated with an improved overall survival in KRAS mutant lung cancer patients (Supplementary Fig. 7b). Subsequently, we stratified the LUAD patient samples based on the expression of these 16 TSGs and found that high expression of these genes is collectively predictive of positive outcomes in KRAS mutant lung cancer patients, but not in patients with KRAS wild-type tumors (Fig. 8d).

## Discussion

Epigenetic changes resulting from abnormal patterns of DNA methylation may lead to altered expression of anti-proliferative and pro-apoptotic genes and promote cancer development[42]. Enzymes mediating these epigenetic alterations have been proposed to cooperate with oncogenes in promoting cancer progression by amplifying and/or complementing their pro-tumorigenic effects[43]. Here, we describe a cooperation between oncogenic KRAS and the epigenetic regulator UHRF1 which appears to be critical for lung tumorigenesis.

Using a combination of genetically engineered mouse models, loss-of-function RNAi screens in primary tumor cells derived from these models, and 3D cultures of human non-small cell lung cancer cell

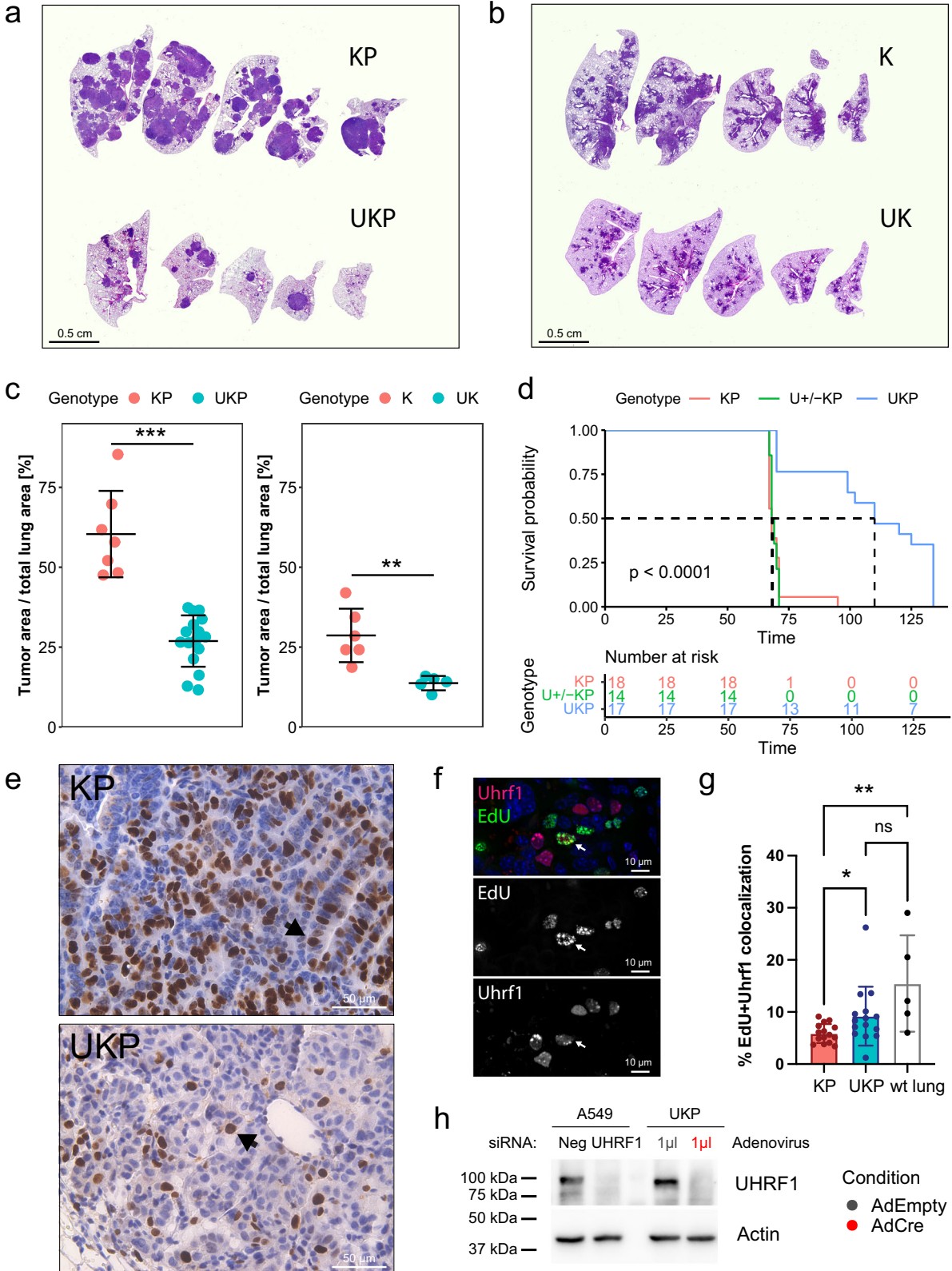

lines we show that UHRF1 plays a key role in KRAS-driven oncogenesis. In vitro, in mouse and human models of NSCLC, *UHRF1* knock-out inhibits 3D growth and leads to apoptosis. In GEM models of lung cancer, loss of Uhrf1 results in decreased tumor number and tumor size and significantly extends survival, even in a highly aggressive model driven by activation of Kras and loss of p53. Indeed, in these mice we could not detect any tumors with complete loss of UHRF1,

suggesting that UHRF1 expression is important for tumor development. In lung cancer patients with tumors harboring a *KRAS* mutation, high UHRF1 expression is anticorrelated with tumor suppressor gene expression and predicts poor patient survival. Collectively our data suggest that UHRF1 contributes to KRAS-driven oncogenesis by inducing hypermethylation of TSG promoter regions leading to their reduced expression. We find that at least 80 lung cancer-specific TSGs

**Fig. 6 | Uhrf1 is essential for tumor growth in a mouse model of Kras-driven lung cancer. a**, **b** H&E images of representative tumor-bearing lungs from UKP and KP mice (**a**) and UK and K mice (**b**). **c** Quantification of tumor burden from the (U)KP (left) and (U)K (right) cohort. Horizontal lines represent mean values with one standard deviation error bars. Significance is calculated using two-sided unpaired Student's *t* test; **\*\****p* = 0.0061, \*\*\**p* = 0.0003. The experiment has been performed twice in (U)KP and once in (U)K mice. **d** Survival analysis of KP (*n* = 18), U + /-KP (*n* = 14) and UKP mice (*n* = 17). Day 0 denotes the day of AdCre administration. Significance calculated using log-rang test, *p* = 3.4454e-07. **e** Immunohistochemistry for Uhrf1 expression in lung sections from representative KP (top) and UKP (bottom) mice. Arrows point to examples of Uhrf1-positive nuclei. Representative

images of *n* = 7 (KP) and *n* = 16 (UKP) animals. **f** EdU and Uhrf1 co-immunostaining of a KP mouse tumor 11 weeks post Cre treated with EdU. Arrows point to a double-positive nucleus. **g** Quantification of Uhrf1 and EdU double-positive cells from EdU treated mice. Number of individual images quantified: *n* = 15 (KP, UKP), *n* = 5 (control no-virus mouse). Bars represent means with standard deviation error bars; \**p* = 0.031, \*\**p* = 0.0086, ns - not significant by Kruskal–Wallis test. **h** Western blot for UHRF1 protein expression in UKP mouse lung cancer cell line infected with the indicated adenovirus and control A549 cells treated with the indicated siRNAs. AdEmpty – empty adenovirus, AdCre – Cre-expressing adenovirus. The experiment was repeated twice. Source data are provided as a Source data file.

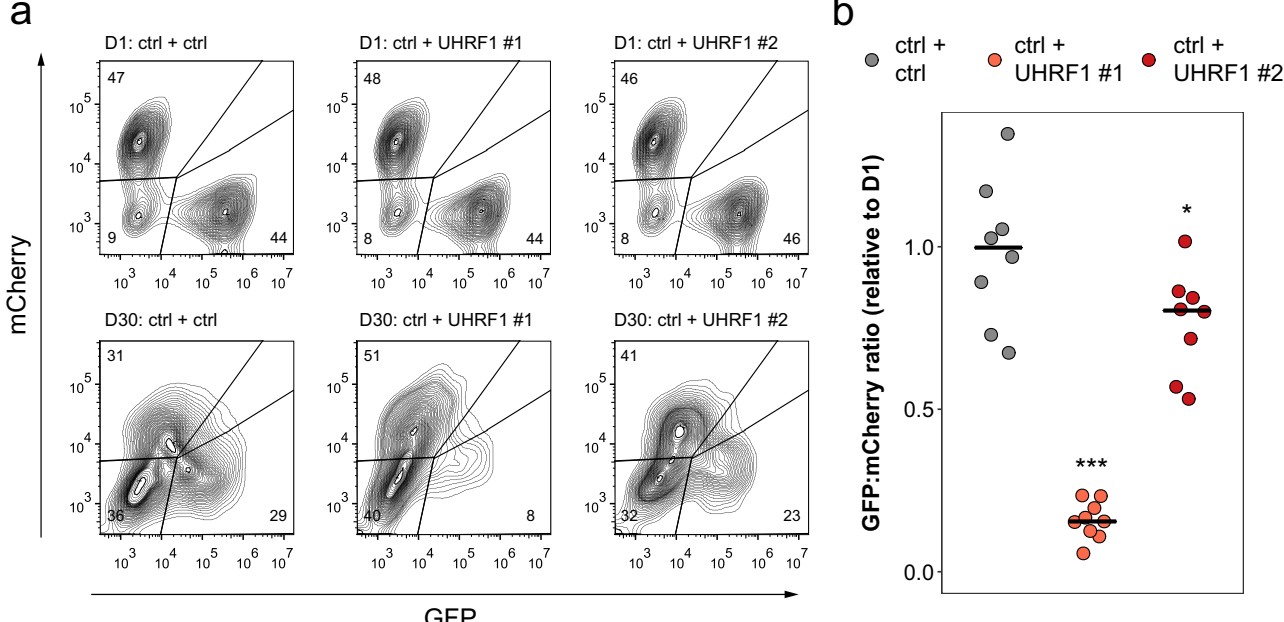

**Fig. 7 | Loss of UHRF1 impairs tumor growth in a xenograft model of human KRAS mutant lung adenocarcinoma. a** Flow cytometry analysis of the in vivo competitive-growth assay in A549 xenografts. Top - day 1 cell populations, bottom – example of endpoint (day 30) tumor cell populations. **b** Fold change in GFP/mCherry ratio between A549 tumor cell populations at endpoint and day 1 (D1) cell populations; *n* = 8 individual tumors assessed for control (ctrl+ctrl) and sgUHRF1-2 (ctrl+UHRF1#2) arms, *n* = 9 individual tumors assessed for sgUHRF1-1 (ctrl+UHRF1 #1) arms. Lines represent means; \**p* = 0.0234, \*\*\**p* = 6.31e-10 by anova followed by Dunnett's test between the indicated UHRF1 sgRNA and the control sgRNA. Source data are provided as a Source data file.

are regulated by UHRF1. Many of these TSGs are suppressors of the WNT/β-catenin pathway or genes involved in the activation of apoptosis. While UHRF1-driven hypermethylation of a limited subset of these TSGs, e.g. *CDH13* or *RASSF1*, has already been shown by others in lung cancer cell lines in vitro[44], here we demonstrate that UHRF1 has a more widespread effect on TSG expression. We hypothesize, that this broad effect across large number of TSGs is what contributes to the role of UHRF1 in lung cancer.

Methylation-driven tumorigenic phenotypes have been best described in colorectal cancer (CRC) and prior studies have also demonstrated that UHRF1 plays a role in mediating these effects in CRC[21,30,45]. Independently, KRAS was also been shown to drive TSG promoter hypermethylation in CRC cells in vitro[46]. Moreover, experiments in CRC cell lines demonstrated that KRAS induces promoter hypermethylation and transcriptional silencing of tumor suppressor genes via ZNF304, a protein which recruits DNMT1-containing complexes to DNA[46]. ZNF304 is not highly expressed in lung tumors and its expression is significantly anti-correlated with UHRF1 expression in lung adenocarcinoma patients carrying a KRAS mutation (Supplementary Fig. 7c), suggesting that UHRF1 may instead mediate this phenotype in KRAS-driven lung cancer. In general, our results indicate that TSGs regulated by UHRF1 in lung cancer are non-overlapping with those proposed to be UHRF1-regulated in CRC, likely due to tissue-specific effects[21,30,45].

The mechanistic link between UHRF1 and KRAS remains to be completely understood. Previously, UHRF1 was predicted to be a transcriptional target of KRAS in lung cancer using the virtual inference of protein activity by enriched regulon analysis (VIPER)[33]. A recent study, using a combination of VIPER and CRISPR screens, also identified UHRF1 as one of the most essential genes in pancreatic cancer[47], suggesting a particular dependence of this largely KRAS-driven cancer on UHRF1. In a mouse model of pancreatic cancer overexpression of Kras G12D was shown to induce Uhrf1 expression[48], while in human PDAC cell lines, KRAS knock-down decreased UHRF1 protein levels[49]. In line with these results, we show that KRAS knock-down leads to reduced UHRF1 mRNA and protein expression in human KRAS mutant lung cancer cell lines (Supplementary Fig. 3c, d and Supplementary Fig. 4e). However, concomitant decrease in Cyclins B1 and D1 and cell cycle analysis by flow cytometry (Supplementary Fig. 3c–e), point to a G1 phase arrest in these cells. UHRF1 expression was previously shown to be cell cycle-dependent and peak in S and G2 phases[31], thus it is possible that the loss of UHRF1 protein is a result of cell cycle arrest in G1 rather than a direct consequence of KRAS depletion. Consistent with this hypothesis, we observed no increase in UHRF1 protein levels in KRAS wild-type cell lines upon transient expression of oncogenic KRAS (Supplementary Fig. 7d). Moreover, UHRF1 expression was shown to be driven by FOXM1 and E2F1[44,50,51], both of which are cell cycle regulated. Of note, loss of

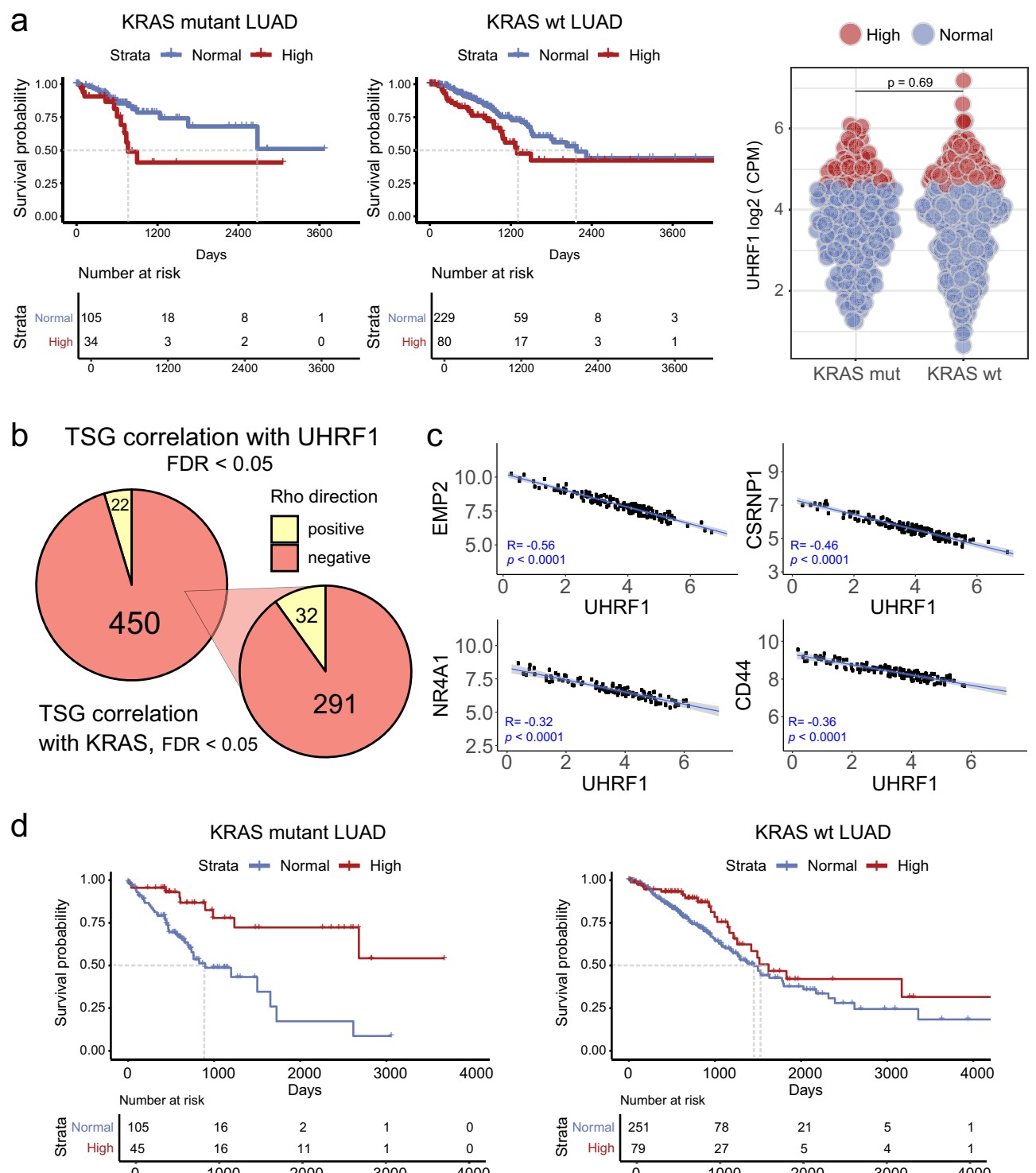

**Fig. 8 | High UHRF1 expression is predictive of poor survival in human KRAS mutant lung adenocarcinoma. a** Kaplan–Meier plots of disease specific survival (DSS) in the lung adenocarcinoma (LUAD) cohort from TCGA. Blue solid line represents the normal UHRF1 expression group (expression <75th percentile) and red solid line represents the high UHRF1 expression group (>75th percentile). Left - KRAS mutant patients, HR = 2.85, *p*-value = 0.01; middle - KRAS wild-type patients, HR = 1.53, *p*-value = 0.092, right – distribution of UHRF1 log2 (CPM) counts by sample and color-coded by high or low group assignments based on 75th percentile. Two-tailed Student's *t* test used to calculate the *p*-value. **b** Left pie chart – Correlation between UHRF1 expression and TSG expression. Right pie chart – Correlation between KRAS expression and expression of 323 TSGs (out of 450 TSGs anticorrelated with UHRF1) which significantly correlate with KRAS expression. Only significantly correlated TSGs (FDR < 0.05) were plotted for both

datasets. Rho Direction: TSGs negatively (neg) or positively (pos) correlated with UHRF1 or KRAS expression. **c** Four examples of TSGs negatively correlated with UHRF1 in the LUAD dataset. Expression represented in log2 (CPM). Correlation coefficient and p-value computed using Spearman rank correlation test. Linear trend lines were generated using a linear model, shaded confidence regions represent CI = 0.95. **d** Kaplan–Meier plots of overall survival (OS) in the lung adenocarcinoma (LUAD) cohort from TCGA. Patient samples were divided based on the expression of 16 TSGs whose expression is anticorrelated with both UHRF1 expression and KRAS expression in the LUAD dataset samples. Blue solid line represents normal expression of the 16 TSGs (expression <75th percentile) and red solid line represents high expression (>75th percentile). Left - KRAS mutant patients, HR = 0.172, *p*-value < 0.001; right - KRAS wild-type patients, HR = 0.794, *p*-value = 0.357.

UHRF1 itself has also been shown to affect cell cycle progression[52], which is supported by our observation that UHRF1 knock-down leads to increased levels of DNA damage and to the accumulation of cells in S phase (Supplementary Fig. 3e, g–j). This effect is also more pronounced in KRAS mutant than in KRAS wild-type NSCLC cells, suggesting that the former may be more sensitive to the accumulation of unresolved DNA double-stranded breaks in S phase.

While UHRF1 expression is generally thought to be regulated in a cell cycle-dependent manner, immunofluorescence imaging of tumors from KP mouse lungs revealed the presence of Uhrf1 in cells at different stages of the cell cycle, including G1 and G0 (Supplementary Fig. 6c, d), which suggests a deregulation of its cell cycle-dependent expression in lung cancer. Therefore, we hypothesize that UHRF1 regulation by mutant KRAS is at least partially cell cycle independent. However, the exact mechanism of this altered expression of UHRF1 will require further investigation.

While multiple proximal effectors of KRAS are well known, how these signaling pathways lead to regulation of gene transcription remains poorly understood. Specifically, the mechanistic link between epigenetic regulators and KRAS-induced oncogenesis has not been fully explored. In a recent study Tew, Durand and colleagues[53] examined the link between oncogenic KRAS and methylation in pancreatic cancer and found that changes in DNA methylation in response to KRAS knock-down are very cell-specific, which results in heterogeneity between cell lines. In line with these observations, we found little overlap in differential methylation between the two lung cell lines subjected to KRAS knock-down, while simultaneously noting a significant increase in CpG methylation variability compared to control cells (Supplementary Fig. 4b). The authors also demonstrated that methylation changes driven by KRAS are independent of canonical MAPK signaling, pointing to other effectors. Our observations suggest that in lung cancer UHRF1 may be a potential effector of KRAS that mediates its role in DNA methylation.

Prior studies show that combining the inhibition of the epigenetic regulators, such as HDAC or BET proteins, with inhibition of MEK or PI3K is efficacious in pre-clinical models of RAS-driven cancers[43,54], supporting the general concept that epigenetic regulators may contribute to KRAS driven oncogenesis. In line with this, here we show that genetic ablation of UHRF1 in combination with pharmacological inhibition of KRAS G12C, MEK, or PI3K results in a synergistically antiproliferative effect in KRAS mutant lung cancer cells in vitro. Thus, our work demonstrates that co-targeting of UHRF1 and KRAS may constitute a viable therapeutic approach for KRAS mutant cancers.

A recent study also found that known chemotherapeutic drugs, including doxorubicin or mitoxantrone, inhibit the binding of UHRF1 to hemi-methylated DNA[55], suggesting that at least some of the effect elicited by these drugs could be a result of UHRF1 inhibition. Moreover, small molecules that specifically target the tandem Tudor domain (TTD) of UHRF1, have recently been described[56]. Thus, in the future targeted chemical inhibition of UHRF1 in cancer may be possible. In addition, a reversible DNMT1-selective small molecule inhibitor was recently reported by Pappalardi and colleagues[36]. Here we shown that in KRAS mutant cancer cells treatment with this inhibitor elicits an effect similar to that of UHRF1 genetic ablation. Therefore, we hypothesize that treatment with DNMT1 inhibitors may constitute an alternative to UHRF1 targeting and may be a potential therapeutic approach for treatment of KRAS-driven cancer. Moreover, recent advances in targeted protein degradation, including molecular glues and PROTACs, demonstrate that these methods are a viable strategy to target genes previously considered undruggable. Our study suggests that UHRF1 may be a good target for a degrader and that patients with tumors carrying KRAS mutations could benefit the most from this approach.

Finally, UHRF1 inhibition may constitute a good combination partner for immune checkpoint inhibitor (ICI) therapy. ICIs have been successfully used for treatment of lung cancer for almost a decade, with durable responses seen in some patients. However, data from the clinic shows that patients with KRAS mutant tumors harboring a KEAP1 co-mutation respond less well to ICIs[57]. Similarly, KRAS G12C patients carrying a KEAP1 co-mutation show lower rates of response to KRAS G12C inhibitor (sotorasib)[3]. This suggests that impaired or low expression of KEAP1 contributes to poorer responses in these patients. Given that high UHRF1 expression was previously shown to lead to decreased KEAP1 levels via hypermethylation of its promoter region, we speculate that co-targeting of UHRF1 together with ICIs or KRASG12Cis, at least in KEAP1 wild-type patients, may lead to improved responses to these therapies by removing the inhibitory effect of UHRF1 on KEAP1. Our data with sotorasib in UHRF1-depleted cells provide the first proof-of-concept for this strategy.

# Methods

## Mice

All the procedures involving mice were are approved by the Institutional Animal Care and Use Committee (IACUC) at UCSF (protocol #AN15761). The mice were housed in the HDFCCC animal facility at UCSF in individually ventilated microisolator cages with automatic watering system purified using reverse osmosis. The cages were on a 12/12 hour light/dark cycle. All feed was pre-irradiated. Temperature was maintained between 20 and 22 °C and humidity at 30–70%. Mice were euthanized by carbon dioxide inhalation followed by cervical dislocation after cessation of vital signs. C57BL/6 Uhrf1$^{fl/fl}$ mice, which harbor loxP sequences flanking exon 4 of the Uhrf1 gene (ENSMUSG00000001228) were a gift from the laboratory of Benjamin Singer (Northwestern University Feinberg School of Medicine, Chicago, IL, USA). Kras$^{LSL-G12D}$ (K, JAX strain no. 008179) and Kras$^{LSL-G12D}$, Trp53$^{fl/fl}$ (KP, JAX strain no. 032435) mice were obtained from Jackson Laboratory, Bar Harbor, ME, USA. To induce tumor formation in the lungs of mice, Cre-expressing adenovirus (University of Iowa) was delivered intranasally into 4–10 week old mice as described previously[58]. Mice were monitored every other day and euthanized at the time of appearance of signs of labored breathing. For endpoint experiments the entire cohort was euthanized when the first mice showed signs of labored breathing (approx. 14 weeks for KP mice and 16 weeks for K mice). For the survival study, each mouse was euthanized when labored breathing was observed. Both male and female mice were used for GEMM studies. NSG mice (JAX Strain no. 005557) were obtained from Jackson Laboratory, Bar Harbor, ME, USA, and bred in the barrier facility at Stanford University or University of California, San Francisco. Both male and female mice were used for xenograft studies. Xenograft tumors were injected when mice were ~8–10 weeks of age. Mice were euthanized before tumor size reached the maximal allowed tumor size of 1 cm³ per flank.

## Primary tumor-propagating cell culture and screening methodology

Primary lung tumor cells from Kras$^{LSL-G12D}$, Trp53$^{fl/fl}$ (KP) mice were cultured in Matrigel as described previously[12]. Before seeding, primary cells were infected with a pool of 100–150 lentiviral pLKO shRNAs composed of 2–5 shRNAs per gene at a multiplicity of infection of <0.5 to ensure single shRNA integration and selected with 1 μg/ml of puromycin 24 h after seeding. In total we screened 115 genes using pooled shRNA libraries composed of 518 shRNAs targeting downstream effectors and upstream modulators of KRAS inferred by the algorithms ARACNe and VIPER[33,34] (Supplementary Data 1), KRAS interactors identified using affinity purification mass spectrometry (AP/MS)[17] (Supplementary Data 2), and controls (Trp53, Tbk1). Sequences of shRNAs used in the screens are listed in Supplementary Data 3. After 7 days of growth, spheroids were dissociated with trypsin into single cells, and half of the culture was re-seeded. The remaining half of each sample was retained for genomic DNA isolation (T0) until secondary spheroids fully formed 7 days later (T1). The integrated pLKO shRNA

was PCR amplified using Ex Taq (Clontech), barcoded, multiplexed and sequenced on an Illumina GAIIx (primer sequences available upon request). Sequencing reads were processed into count files in R (v. 3.1.1) using the edgeR package (v. 3.6.8) and analyzed using generalized linear models with edgeR using a time course design to compare the initial (T1) and final (T2) time points to derive statistical significance using the likelihood ratio (LR) test[59]. Log2 fold change (Log2FC) of each shRNA was measured between T2 and T1 samples. Log2FC, *p*-values and *p*-values corrected for false discovery rate are shown in Supplementary Data 4.

## Cell lines

Human NSCLC cell lines (NCI-H1437 #CRL-5872, NCI-H1568 #CRL-5876, NCI-H1650 #CRL-5883, NCI-H1975 #CRL-5908, NCI-H460 #HTB-177, NCI-H1792 #CRL-5895, NCI-H2009 #CRL-5911, NCI-H23 #CRL-5800, NCI-H358 #CRL-5807, A549 #CCL-185), HBEC cell lines (NL20 #CRL-2503, BEAS-2B #CRL-9609), and HEK 293 T #CRL-3216 cells were obtained from the ATCC. All NSCLC cell lines were maintained in RPMI 1640 (Corning, #15-040-CV) supplemented with 10% FBS and 1% penicillin–streptomycin-glutamine (Gibco, #10378016). NL-20 cells were cultured in Ham's F12 medium (Gibco, #11765054) supplemented 2.7 g/L glucose (Sigma, #G8270), 1% penicillin–streptomycin-glutamine, 1x MEM Non-Essential Amino Acids (Gibco, #11140050), 1x ITSE (InVitria, #777ITS032), 10 ng/ml EGF (Humanzyme, #HZ-7012), 500 ng/ml hydrocortisone (Sigma, #H0888), and 4% FBS. BEAS-2B cells were grown in BEGM™ medium (Lonza, CC-2540B) on plates coated with 0.01 mg/mL fibronectin (Corning, #354008), 0.03 mg/mL bovine collagen type I (PureCol, #5005) and 0.01 mg/mL bovine serum albumin (Sigma, #A7030). All cell lines were authenticated using the Human 9-Marker STR Profile test provided by IDEXX BioResearch (completed on 2.2.2018) and regularly tested for mycoplasma contamination. Murine LKR10 cells were a gift of Julien Sage (Stanford School of Medicine, Stanford, CA, USA) and were grown in DMEM supplemented with 10% FBS and 1% penicillin–streptomycin. To obtain Cas9-expressing cell lines, cells were transduced with a spCas9 lentiviral vector with a blasticidin selection marker (Addgene no. 52962), and selected with blasticidin (5–10 μg/ml). Blasticidin-resistant polyclonal cell populations were tested for their Cas9-cutting efficiency by lentiviral infection with pMCB306[60], a self-GFP-cutting reporter that expresses GFP and an sgRNA against GFP.

## Inhibitors

Pharmacological inhibitors of MEK1/2 (Trametinib/GSK1120212), PI3K (Copanlisib/BAY 80-6946), and DNMT1 (GSK3685032) were acquired from Selleckchem. KRAS G12C inhibitor (Sotorasib/AMG-510) was purchased from MedChemExpress.

## Vectors

SgRNAs against UHRF1 or control safe-targeting sgRNAs were purchased from Integrated DNA Technologies (IDT) and cloned into the MCB306[60] or KH91 vector (modified MCB306 with a mCherry instead of GFP). All sgRNA sequences are listed in Table 1. Lentiviral vectors were produced by transfecting 293T cells with the lentiviral vectors and delta8.2 and VSV-G packaging plasmids. Lentivirus-containing supernatant was collected, filtered, and applied directly to cells for infection at an MOI lower than 1.

## Lentivirus infection and indel efficiency analysis

Cas9-expressing cells were seeded into 24-well plates and 24 h later infected with lentiviral vectors expressing a safe-cutting control sgRNA or one of the two UHRF1-targetting sgRNAs (sgRNA #1 or #2). Puromycin (2 μg/ml) was added to the cells 48 h post infection and cells were selected for 2 days. Following puromycin selection the cells were split, counted and seeded for in vitro assays. For indel efficiency analysis the cells were seeded into six-well plates, cultured for 5 days and

## Table 1 | sgRNA sequences in lentiviral vectors

| Vector | Target | sgRNA sequence | Reporter |
|---|---|---|---|
| UHRF1_1 | UHRF1 | GGACAGCGAGTCCACCGTGT | GFP |
| UHRF1_2 | UHRF1 | GTAGAGTTCCCGCGCCGTCC | GFP |
| pGH119 | safe | GTCAGTTCCTATGTGGCA | GFP |
| pGH119 | safe | GTCAGTTCCTATGTGGCA | mCherry |

subsequently collected for DNA extraction. Indel analysis was performed using TIDE[61] and analyzed using the online tool: http://shinyapps.datacurators.nl/tide/.

## Sphere formation assay

For anchorage-independent sphere growth the cells were seeded into 24-well ultra-low attachment plates (Corning, #3473) in 2 ml of complete medium supplemented with 0.05% methylcellulose (20,000 viable cells per well). The spheres were allowed to form for 9–20 days (depending on the cell line). GFP-positive spheres were imaged using the Leica DMi8 fluorescence microscope. Sphere size and number were quantified using ImageJ (NIH, Bethesda, Maryland, USA). For sphere growth assays in the IncuCyte S3 (Essen Bioscience) cells were seeded into complete medium supplemented with 1.5% methylcellulose (400–1000 cells/well) in 96-well ultra-low attachment plates (Corning, #3474). Each condition was seeded in triplicate. For inhibitor treatment assays, drugs were added 48 h after seeding. Cell area mask was calculated on the green channel to include only GFP-positive cells.

## Colony formation assays

For 2D colony formation assays cells were trypsinized, counted and 10'000 viable cells were seeded in complete medium into each well of a 6-well plate. After 10 days colonies were stained with 0.2% methylene blue and quantified using ImageJ (NIH, Bethesda, MD, USA).

## Proliferation assays

Short term 2D cell viability assays were performed using the IncuCyte S3 (Essen Bioscience). Five days post infection with lentivirus the cells were trypsinized, counted and 500–1000 viable cells were seeded in 200 μl of puromycin-containing medium into each well of a 96-well plate. Each condition was seeded in triplicate. Cells were imaged every 12 h for the duration of the experiment. Cell area mask was calculated on the green channel to include only GFP-positive cells.

## Western blotting

Cells were washed in ice-cold PBS, then lysed in Triton buffer (20 mM Tris (pH7.5), 150 mM NaCl, 1 mM EDTA, 1% Triton X-100, 10 mM NaF, 2.5 mM sodium pyrophosphate, 1 mM b-glycerophosphate, 1 mM Na3VO4 supplemented with protease inhibitors (Roche). Cleared supernatants were subjected to protein quantification by Bradford (BioRad) or BCA kit (Pierce). Proteins were resolved by SDS-PAGE, transferred to PVDF membranes, and blocked in 5% non-fat dry milk or 5% BSA in TBST. Samples were incubated overnight at 4 °C with the following primary antibodies at the indicated dilutions: UHRF1 (sc-373750, Santa Cruz Biotechnology, 1:100), KRAS (Op24, Millipore, 1:100), Cyclin D1 (ab134175, Abcam, 1:10,000), actin (A5316, Sigma, 1:500), Cyclin B1 (ab32053, Abcam, 1:5000), p-ERK1/2 (#4370, Cell Signaling, 1:1000), ERK1/2 (#4695, Cell Signaling, 1:1000), p-AKT (#13038, Cell Signaling, 1:1000), AKT (#75692, Cell Signaling, 1:1000), and GAPDH (ab9485, Abcam, 1:5000). Proteins were analyzed by ChemiDoc XRS System (Bio-Rad) and when necessary, quantification was performed using ImageJ (NIH, Bethesda, Maryland, USA).

## Apoptosis and cell cycle analysis

Cells were seeded into six-well plates. After 3–5 days medium was collected and stored on ice. Cells were trypsinized and transferred to

collected medium. Number of total cells was counted and $1 \times 10^6$ cells were transferred to new tubes. For cell cycle analysis the cells were washed in cold PBS, fixed by adding ice-cold 70% ethanol drop-wise with gentle vortexing and fixed for 30 min at −20 °C. After fixation, cells were washed in cold PBS, then incubated in the dark for 30 min at 37 °C in 500 μl of staining solution containing PBS, 200 μg/ml RNAse A (Qiagen, 17,500 U, #19101) and 40 μg/ml propidium iodide (Invitrogen, #P3566). After incubation the cells were centrifuged, resuspended in 500 μl of PBS and analyzed by flow cytometry on Accuri C6 (BD Biosciences). For apoptosis analysis the cells were stained with APC Annexin V Apoptosis Detection Kit with propidium iodide (BioLegend #640932) following manufacturer's instructions and analyzed by flow cytometry on Accuri C6 (BD Biosciences).

### Competition assay in tumor xenografts

A549-iCas9 cells were infected with the indicated sgRNA vectors and puromycin selection was applied for 7 days. Stably infected cell populations were trypsinized and transferred to 15 cm dishes to expand. Three days prior to implantation the cells were treated with doxycycline (1 μg/ml). On the day of the implantation (D1) cells were trypsinized, counted and analyzed by flow cytometry. Cells expressing mCherry (safe control sgRNA vector) and GFP (UHRF1-targeting sgRNA vectors) were analyzed by flow cytometry on Accuri C6 (BD Biosciences), pooled 1:1, centrifuged and re-suspended in a mixture of serum-free DMEM (Corning, #15-017-CV) and Matrigel (Corning, #356237). Cells were injected subcutaneously into both flanks of NSG mice ($2 \times 10^6$ cells per flank, 3 mice per condition). Thirty days after implantation, tumors were dissected, chopped and dissociated into a single-cell suspension using PBS supplemented with Collagenase/Dispase (Roche, #11097113001) and DNAse I (Worthington, #LS002006) at 37 °C for 30 min with agitation. Digested samples were passed through a 40μm filter, resuspended in complete DMEM medium and centrifuged. Cell pellets were incubated for 1 min in RBC buffer (155 mM NH4Cl, 12 mM NaHCO3, 0.1 mM EDTA) to remove red blood cells, resuspended in complete DMEM and centrifuged. Resulting cell pellets were resuspended in PBS with 1% serum and analyzed by flow cytometry on Accuri C6 (BD Biosciences). The ratio of GFP to mCherry fluorescence in tumor samples was normalized to D1 cell populations and represented as fold-change between these two time points.

### Histology and immunostaining

Tissue specimens were fixed in 10% buffered formalin for 24 h and stored in 70% ethanol until paraffin embedding. Paraffin-embedded tumors were sectioned into 5μm-thick slices and used for hematoxylin and eosin (H&E) staining, immunohistochemistry (IHC) or immunofluorescence (IF). For IHC and IF sections were deparaffinized with xylene and ethanol and antigen-retrieved in citrate buffer. Blocking and antibody dilutions were made in 5% normal goat or rabbit serum in TBST and incubated overnight at 4 °C in a humidified chamber. For IF on cell lines, after 72 h of siRNA transfection, cells were fixed in 4% formaldehyde for 15 min at room temperature and subsequently washed three times in PBS followed by a 1 h incubation in the blocking solution. The following antibodies were used (at indicated dilutions): Uhrf1 (sc-373750, Santa Cruz Biotechnology 1:50), Ki67 (ab15580, Abcam, 1:1000) and pH2AX (Cell Signaling, #9718, 1:200). For IHC biotinylated secondary horse anti-mouse IgG biotinylated antibody (Vector Laboratories, BA-2000, 1:1000) and avidin-biotin Vectastain Elite ABC kit (#PK-6100) were incubated for 30 min at room temperature. Signal was developed with DAB peroxide substrate (Abcam, ab94665) per manufacturer's instructions and counterstained with hematoxylin. For IF goat anti-rabbit AF488 secondary antibody (ThermoFisher Scientific, A-11008, 1:200), goat anti-mouse AF647 secondary antibody 1:200 (ThermoFisher Scientific, A-21235, 1:200), and Vectashield mounting medium with DAPI (Vector Laboratories, H-1200) were used to visualize Ki67, UHRF1 and nuclei, respectively. Goat anti-mouse

AF488 secondary antibody (ThermoFisher Scientific, A-11029, 1:200) and goat anti-rabbit AF647 secondary antibody (ThermoFisher Scientific, A-21245, 1:200), and ProLong Gold Antifade mounting medium with DAPI (ThermoFisher Scientific, P3691) were used to visualize UHRF1, pH2AX and nuclei, respectively. Images were taken on DMi8 fluorescence microscope (Leica) using ×20, ×40, or ×100 objectives.

### Tumor burden quantification

CellProfiler[62] is an open-source software for measuring and analyzing cell images. We designed a custom Cell Profiler pipeline (software version 3.1.9) to automatically quantify tumor burden in murine lung H&E samples. The pipeline has four stages. In the first stage the H&E images are imported and classified as color images. The second stage creates separate grayscale images from color images stained with light-absorbing dyes, allowing the separation of the two stains from the background. This stage generates two independent images corresponding to the tumor lesions and normal lung tissue. During the third stage, the pipeline produces binary images based on a pre-selected threshold for both the tumor area and the normal tissue area, and converts images into segmented objects. For thresholding, the Global Otsu two-classes method is used. The final stage measures the area occupied by the specific objects (lung lesions and normal tissue) and calculates the percentage of tumor area. The full pipeline is available upon request.

### In vivo proliferation analysis

Mice were injected i.p. with 50 mg/kg of EdU in PBS and lungs were harvested 16 h after injection. EdU and UHRF1 protein expression was evaluated using the Click-IT Edu Alexa Fluor 488 Imaging kit (C10337, Invitrogen) in formalin-fixed and paraffin-embedded lung tissue sections. After paraffin removal, endogenous peroxidase activity was inhibited with 3% hydrogen peroxidase in PBS for 10 min at RT. Antigen retrieval was carried out by boiling samples in Sodium Citrate (10 mM, pH 6) for 15 min. Sections were blocked using 5% Goat serum for 1 hour and incubated with the Click-IT EdU reaction cocktail following manufacturer's protocol. Then, sections were washed with 1X TBST and re-blocked during 30 min before incubation with mouse anti-UHRF1 (1:50, sc373750, Santa Cruz Biotechnology) at 4 °C overnight. Detection was conducted with the M.O.M Immunodetection kit (BMK-2202 Vector laboratories) following manufacturer's instructions and using Streptavidin Alexa Fluor-647 (1:200, S21374, Invitrogen) to detect UHRF1 staining. Finally, sections were counterstained with Hoechst and mounted for fluorescence microscopy.

### Quantification EdU/Uhrf1 co-localization

To quantify the percentage of cells expressing both UHRF1 and Edu (i.e. "co-localization") we developed a custom pipeline using Cell Profiler (software version 3.1.9)[63]. This pipeline uses grayscale images as input and separates them by fluorescent channel (FITC-EdU, Cy5-UHRF1 and Hoechst-nuclei). Subsequently the pipeline identifies biological objects of interest for each channel, which correspond to those cells positive for each staining, according to the typical diameter of the object in pixel units. At this stage, a pre-selected threshold using the Otsu method is also applied to the images. Subsequently, using the "RelateObjects" module, we can assign relationships between objects to identify double expressing cells though the centroid method. On the one hand, we identify double Hoechst+ EdU+ cells and Hoechst+ UHRF1+ cells to later relate Hoechst + /UHRF1+ cells ("parent") to Hoechst + /EdU+ cells ("child"). These relationships allow to obtain measurement values for those child objects associated with specific parent objects. The full pipeline is available upon request.

### Extraction of nucleic acids

For qRT-PCR total RNA was isolated using the RNeasy Mini Kit (Qiagen, #74104) with QIAshredder homogenization (Qiagen, #79654). For

**Table 2 | Primer sequences for qRT-PCR**

| Gene | Forward primer | Reverse primer |
|---|---|---|
| CD44 | GAGCAGCACTTCAGGAGGTT | TGGTTGCTGTCTCAGTTGCT |
| NR4A1 | GTTCTCTGGAGGTCATCCGCAAG | GCAGGGACCTTGAGAAGGCCA |
| ZNF185 | GGAGACACAGGCACCGTTTA | GCACTCGATCCAAATTGCCC |
| GABARAP | CTCTGAGGGCGAGAAGATCC | TCCAGGTCTCCTATCCGAGC |
| UHRF1 | GACAAGCAGCTCATGTGCGATG | AGTACCACCTCGCTGGCATCAT |
| B2M | CGCTACTCTCTCTTTCTGGC | GACTTTCCATTCTCTGCTGG |

sequencing and DNA methylation DNA and RNA were co-extracted using the AllPrep Kit (Qiagen, #80204) with QIAshredder homogenization (Qiagen, #79654). DNA was quantified using the Nanodrop 2000 (Thermo Fisher) and the QuBit High Sensitivity dsDNA assay (Thermo Fisher, #Q32851). RNA was quantified using Nanodrop 2000 (Thermo Fisher) and QuBit High Sensitivity RNA assay (Thermo Fisher, #Q32852). Prior to sequencing, RNA integrity was quantified using High Sensitivity RNA ScreenTape (Agilent, #5067-5579) on a TapeStation 4200 (Agilent).

**Quantitative real-time RT-PCR analysis**

RNA was converted to cDNA with the Maxima First Strand cDNA Synthesis Kit (Thermo Fisher, #K1641) following manufacturer's instructions. Reverse transcriptase reactions were performed using 1 μg of RNA. Each sample was analyzed in triplicate. We developed quantitative SYBR green PCR assays for the 5 human genes involved in this study. Primers for each target have been purchased from Integrated DNA Technologies (IDT) and their sequences are listed in Table 2. Real-time PCR amplification was performed in the C1000 Touch Thermal Cycler (BioRad) using the PerfeCTa SYBR Green FastMix Reaction Mix (QuantaBio, #101414-276). Beta-2 microglobulin was used as an endogenous control and differences between samples were calculated using the $2^{-\Delta Ct}$ (delta CT) method.

**Gene expression analysis**

RNA-seq libraries were made using the TruSeq Stranded mRNA Kit (Illumina, RS-122-2101) with an input of 500 ng in accordance with the manufacturer's instructions. All manufacturer controls were used in preparation of the libraries. Libraries were quantified using the High Sensitivity D1000 ScreenTape (Agilent, 5067-5584) on the TapeStation 4200 (Agilent). Sequencing was performed on an Illumina NovaSeq 6000 system using chemistry for 150 bp paired-end reads at the Center for Advanced Technology at UCSF. RNA-seq FASTQ data were preprocessed using HTStream (UC Davis, https://github.com/s4hts/HTStream) and aligned to the GRCh38 reference genome v37 (https://www.gencodegenes.org/human/) using STAR (2.5.3a)[64] to obtain gene-level counts. Gene normalization and expression was calculated as log2(CPM + 1) using using edgeR (3.28.1)[65]. This data can be accessed under the SuperSeries accession number GSE198450, specifically denoted as GSE198289.

**Methylation analysis**

The EPIC methylation array was performed at the Vincent J. Coates Genomics Sequencing Laboratory at UC Berkeley. Downstream analysis was conducted with R (3.6.2) under Debian 10 OS[66]. IDAT files were taken as input, preprocessed, and normalized using the Minfi (1.32) package. Quality control/preprocessing included removal of low-quality probes and probes on problematic positions such as probes that were on multi-mapped, SNPs, or cross-reactive sites. Following initial filtering and quality check, the data was normalized using the quantile normalization method. Batch effects and cell-type heterogeneity (EpiDISH, 2.2.2) were assessed. Both Beta and M-values were then generated. As recommended, all statistical analyses were generated with M-values while plots were displayed as Beta-values[67].

Additional annotations such as promoter region and island prediction were all acquired with annotations data packages from Bioconductor, IlluminaHumanMethylationEPICanno.ilm10b4.hg19 (0.6). Unsupervised t-SNE was performed on the top 10,000 most variable CpGs (on the basis of median absolute deviation). Single differential methylation probe (DMP) analysis was performed using the linear modeling provided by the limma package (3.42.2)[68] and differential variable probe (DVP) was performed with the missMethyl (1.20.4) package[69]. The DMRcate (2.0.7) packaged was then used to determine differential methylated regions (DMR)[70]. The microarray data can be accessed under the SuperSeries accession number GSE198450, specifically denoted as GSE198446.

**Minipool CRISPR/Cas9 screening**

To design the CRISPR/Cas9 library we selected 80 tumor suppressor genes hypomethylated and transcriptionally upregulated in UHRF1-depleted A549 and H358 cells, 10 positive control genes, as well as UHRF1 and DNMT1 (Supplementary Data 9). The resulting custom pooled CRISPR/Cas9 library contains 483 sgRNAs targeting 92 coding genes (5 sgRNAs per gene) and 26 negative control "safe" sgRNAs targeting nonfunctional regions of human genomic regions[10]. The sgRNA library was synthetized as a single oligo pool (Twist Bioscience), PCR-amplified and cloned into the lentiviral vector pMB160, derived from pLG20 (gift from Luke Gilbert, UCSF). This plasmid is identical to pLG20, except for the addition of a 497 bp stuffer sequence between the cloning sites BstXI and BlpI. This stuffer facilitates high-grade purification by separating double-digestion products from incompletely digested vectors. We followed the cloning protocol as previously described (https://weissman.wi.mit.edu/crispr/). Subsequently, the lentiviral pMB160 plasmid was transfected into HEK293T cells to produce lentiviral pools. The screen was performed in A549 stably expressing Cas9 cells, which were infected with the lentiviral library at low MOI (0.25 – 0.35) to ensure that every clone receives one unique sgRNA. Three days after infection, cells were placed under puromycin selection (1 μg/ml) and expanded for an additional 3 days before transfection with control or UHRF1-targeting sgRNA. The cells containing the library were transfected with an UHRF1 oligo sgRNA (GGACAGCGAGUCCACCGUGU) or a negative control oligo sgRNA (scramble sgRNA#1 from Synthego) using Lipofectamine RNAiMAX following manufacturer's instructions. Cells were then expanded for 4-5 days, then split. At this point cells were collected for timepoint 0 (T0). During the screen, the pooled libraries were maintained at 1000 cells per sgRNA. A549-Cas9 cells (sgNeg and sgUHRF1) containing the CRISPR library were cultured for 18 days, at the end of which end point cell pellets were collected. DNA was extracted using Qiagen's Blood Maxi Kit, sgRNA cassettes were PCR-amplified from genomic DNA (T0 and end points) using NEB Q5® DNA Polymerase in order to add deep sequencing adapters and sample barcodes. Finally, sgRNA composition was analyzed by deep sequencing using HiSeq4000 SE65 technology (Illumina). MAGeCK software version 0.5.4[71] was used to calculate the phenotype of sgRNAs and to compare frequencies between end point and T0 for each condition (sgNeg and sgUHRF1 cells containing the CRISPR library). Results of the analysis were included in Supplementary Data 10. The CRISPR screen data has

been deposited to NCBI GEO under the accession number GSE198450, specifically denoted as GSE233401.

## Survival analysis

Standardized clinical survival end-points for TCGA (LUAD dataset: https://gdac.broadinstitute.org/runs/stddata_2016_01_28/data/LUAD/20160128/gdac.broadinstitute.org_LUAD.Merge_rnaseqv2_illuminahiseq_rnaseqv2_unc_edu_Level_3_RSEM_genes_data.Level_3.2016012800.0.0.tar.gz), consisting of DSS, OSS and OS were downloaded from a dataset previously curated from Liu et al.[72]. This set contains survival end-points for overall survival (OS), disease-specific survival (DSS), disease-free interval (DFI) and progression-free interval (PFI). Somatic mutation for the LUAD data set was acquired from the UCSC Xena public repository. Samples with non-silent KRAS mutation(s) were assigned to the KRAS-positive (KRAS mut) group and all others as KRAS wild-type (KRAS wt) group. Samples with KRAS silent mutations were excluded from the analysis. Kaplan–Meier, multivariate Cox hazard regression and visualization were completed using the survminer (0.4.6.999) and survival (3.1.11) packages. For the multivariate Cox regression analysis, we adjusted for age of diagnosis, gender, and cancer stage. We grouped samples as Normal vs High expression based on the quantile of the UHRF1 gene expression: normal is <75th percentile and high is >75th percentile.

## Correlation analysis

Spearman correlation and p-value (Benjamini–Hochberg false discovery rate) were computed for UHRF1 against every gene in the tumor suppressor gene list. Moreover, the DGCA (1.0.2) package was used to determine differential correlations between KRASwt and KRASmut subsets[73]. Specifically, we looked for genes that showed statistically different correlation with UHRF1 in the KRASwt vs the KRASmut gene sets.

## Pathway and tumor suppressor gene analysis

Gene set enrichment analysis (GSEA)[74] on genes differentially expressed in UHRF1 knock-down cells was performed using gene sets from KEGG, WikiPathways, Biocarta, Hallmark and GO. The most significantly up- and downregulated pathways (FDR < 0.05) were visualized using Enrichment Map[37]. The activity of known cancer-related pathways was calculated from gene expression data using PROGENy[38]. For pathway analysis on the EPIC methylation array data we used methylGSA (1.4.9) to perform GSEA[75]. Results for GO and KEGG and RAECTOME were exported to Cytoscape (3.8.2) and clueGO (2.5.9)[76] to compute kappa statistic and aggregate the various pathways. Lung cancer specific list of tumor suppressor genes (TSGs) was obtained from TSGene database (The University of Texas Health Science Center at Houston, https://bioinfo.uth.edu/TSGene/). In addition, 12 TSGs from the Cancer Genome Atlas Research Network publication on lung adenocarcinoma[41], not present in the set from TSGgene set, were also included for a total of 537 genes. The lung cancer cell line genetic dependencies for KRAS, UHRF1 and MYC estimated from shRNA screens[8,77] using the DEMETER2 (v5) model were downloaded from the DepMap website (https://depmap.org/portal/download/).

## Statistics

Where indicated in the figure legends statistical significance was calculated using either two-sided unpaired Student's t test between two samples or anova followed by Dunnett's test for multiple comparisons with a control. Where applicable p-values were corrected for multiple-testing using the Benjamini–Hochberg procedure. For gene expression analysis statistical significance was calculated with a linear model and an empirical Bayes moderation of the standard errors, which results in a moderated t-statistic as described in limma[68]. For differential methylation, we used limma where the p-values for gene expression are calculated with a linear model and an empirical Bayes moderation of the standard errors, which results in a moderated t-statistic. For GSEA p-values for gene set enrichment analysis is derived from permutation test and a random walk process as described by A. Sergushichev[78]. For statistical significance in CRISPR screens p-values have been calculated as described by Li et al.[71]. False discovery rate (FDR) correction of the p-values was applied where indicated using the Benjamini–Hochberg procedure.

## Reporting summary

Further information on research design is available in the Nature Portfolio Reporting Summary linked to this article.

## Data availability

The methylation microarray, RNA sequencing and CRISPR screen data has been deposited to NCBI GEO under one superseries with the accession number GSE198450. The source data underlying all main figures and Supplementary figures are provided as a Source Data file. Publicly available datasets used in the study are listed in Supplementary Data 12. All data are available in the main article, Supplementary Information files, and Source Data files. Source data are provided with this paper.

## Code availability

Scripts, codes, and selected publicly available datasets for multiomic analysis, which includes the import of raw sequences, preprocessing, filtering, statistical analysis, and final reports are publicly accessible at a public repository: https://github.com/ahdee/Kostyrko_2023. Custom Cell Profiler pipelines used to quantify tumor burden in IHC images of mouse lung sections and to measure EdU/Uhrf1 co-localization in IF images are available upon request.

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

## Acknowledgements
E.A.S.C. and P.K.J. were funded by U01CA199216 from the National Cancer Institute. E.A.S.C. was funded by a grant from the Stand Up to Cancer Foundation. A.C. was funded by R35CA197745 from the National Cancer Institute; and S10OD012351 and S10OD021764, both from Office of the Director, NIH. K.K. was supported by SNSF Postdoc Mobility Grants P2LAP3_164922 and P300PB_174377. D.R.S. was funded by PHS Grant CA09302 (National Cancer Institute). K.D.M. was funded by Tobacco Related Disease Research Program (TRDRP). M.R. was funded by a fellowship provided by Fundación Ramón Areces. Uhrf1^fl/fl mice were a gift from the laboratory of Benjamin D. Singer (Northwestern University Feinberg School of Medicine, Chicago, Illinois, USA). We thank Kathryn Helmin for assistance with the Uhrf1^fl/fl mice and Sarah Pyle for artwork for Fig. 1a.

## Author contributions
K.K., D.R.S., A.C., P.K.J., and E.A.S.C. conceived and designed the study. K.K., D.R.S., M.R., and M.R.K. performed the experiments with assistance from K.D.M., P.T.D., and S.G.L. K.K., D.R.S., A.G.L., M.R., M.R.K., and J.B. analyzed the data. K.K., M.R., and A.G.L. prepared the figures. K.K., A.G.L., and E.A.S.C. wrote the manuscript. P.K.J., A.C., and E.A.S.C. acquired funding for the study. All authors have read and approved the manuscript.

## Competing interests
A.C. is a founder, equity holder, and consultant of DarwinHealth Inc., a company that has licensed some of the algorithms used in this manuscript from Columbia University. Columbia University is also an equity holder in DarwinHealth Inc. The remaining authors declare no competing interests.
