## [Peer Review File · Nature Communications]

UHRF1 is a mediator of KRAS driven oncogenesis in lung adenocarcinomaREVIEWER COMMENTS

Reviewer #1 (Remarks to the Author):

In this study, Kostyrko and colleagues characterized the role of the epigenetic regulator UHRF1 in KRAS mutant lung cancer. UHRF1 is a RING domain E3 that, in partner with the DNA methyltransferase DNMT1, plays a critical role for maintaining CpG methylation. In addition, UHRF1 has been implicated in the DNA damage response through its interaction with several DNA damage repair proteins. Existing literature suggests that UHRF1 is an essential gene that is required for cell proliferation and stem cell self-renewal. UHRF1 expression is higher in proliferating cells and is often dysregulated in cancer. In the latter context, down-regulation of UHRF1 has been shown to result in the up-regulation of multiple tumor suppressor genes (TSGs) that are repressed by hyper-methylation in cancer cells. UHRF1 knockdown can also induce DNA damage in cancer cells and sensitize them to chemotherapy agents. While signaling pathways downstream of the mutant KRAS oncoprotein has been extensively characterized, particularly in the context of cell proliferation and cell survival through MAP kinase and PI3 kinase signaling, KRAS mutation has also been shown to induce epigenetic changes and DNA damage in cancer cells. How these latter changes influence the behavior of KRAS mutant cells is not fully understood. The authors identified UHRF1 from an RNAi screen comparing genetic dependencies of mouse Kras mutant cancer cells cultured in 3D spheroid vs. 2D monolayer conditions. UHRF1 knockdown showed greater impact on Kras mutant cell proliferation in spheroid culture. The authors also generated UHRF1 CRISPR KO cell lines in human lung cancer cell lines with or without KRAS mutation. In these cells, partial deletion of UHRF1 had variable effect on cell proliferation in 2D, and KRAS mutant cells appeared to be more sensitive to UHRF1 deletion due to higher levels of apoptosis. The authors further validated these findings by generating mouse models of conditional Uhrf1 KO in the Kras/Tp53 background, and showed that Uhrf1 deletion decreased the size and frequency of lung tumors. At the molecular level, the authors showed that UHRF1 deletion caused both hypo-methylation and changes in the expression of a relatively large number of genes. Overlapping methylation and gene expression identified a small set of ~80 TSGs that are upregulated upon UHRF1 loss.

The novelty of this work is that it is the first study to detail the collateral dependency (or synthetic lethality) between UHRF1 and KRAS in lung cancer and the associated molecular changes. The mouse conditional KO models for Uhrf1 provides strong *in vivo* evidence for a functional interaction between Uhrf1 and KRAS. Previously, it has been shown that UHRF1 is required for the proliferation of pancreatic cancer cell lines (most of which are KRAS mutant), and in some lines the underlying mechanism is the repression of the TSG KEAP1 to sustain NRF2 activity (PMID: 26497117). This current study suggests that UHRF1 is a KRAS-dependent vulnerability in lung cancer and a potential target. While UHRF1 itself might not be immediately druggable, its binding partner, DNMT1, is a druggable target. The weakness of this study is that it lacks a clear mechanistic explanation of the synthetic lethal relationship between UHRF1 and KRAS.

Major points:

1. For the primary RNAi screen, the authors compared Kras:Tp53 mutant primary lung tumor cells cultured in Matrigel-based 3D culture vs. a Kras mutant (but not Tp53 mutant?) cell line cultured in standard media in 2D culture. The use of different cell lines and different culture media under these two conditions could complicate the result by introducing additional variables. As collateral dependency genes are often context dependent, their essentiality might be influenced by media and genetic background of the cells.
2. UHRF1 is likely to be an essential gene in normal cells due to its role in CpG methylation and cell cycle progression. The authors showed that partial loss of UHRF1 function, through either siRNA knockdown or incomplete CRISPR KO, causes greater dependencies in 3D culture. Why cancer cells in spheroid culture are more sensitive to the perturbation of UHRF1 than those in 2D culture was not well articulated. If I understood the methods correctly, gene expression and gene methylation studies were conducted using cells in 2D culture, and it is unclear whether these changes are also observed in 3D culture.
3. The authors presented evidence that UHRF1 knockdown is more deleterious to KRAS mutant cells compared to KRAS WT cells. Do the KRAS WT cells used in this study harbor EGFR mutation? Mechanistically, it was not clear as to why UHRF1 depletion is particularly toxic in KRAS mutant cells. Consistent with previous reports, a subset of ~80 TSGs were up-regulated in UHRF1-deleted

cells. It would be useful to test whether one or more such TSGs are responsible for the phenotype, for example, by carrying out double KO experiments.

4. The authors showed that UHRF1 KO cells had higher levels of apoptosis and elevated Trail expression. Trail signaling is known to induce apoptosis. It would be useful to investigate whether cell death is caused by Trail up-regulation or a Trail-independent mechanism. Also, do the UHRF1 KO cells exhibit higher levels of DNA damage in KRAS mutant cells compared to KRAS WT cells?

5. Genetically it has been shown that, with regard to DNA methylation, UHRF1 loss can phenocopy DNMT1 loss. To further explore the translational potential of this work, the authors could attempt to target DNMT1 using inhibitors (for example, GSK3685032) and evaluate whether DNMT1 inhibition is effective at attenuating the growth of KRAS mutant lung cancer cells in vitro and in vivo. In addition, if the primary mechanism of UHRF1 depletion is TSG upregulation, it might be expected that UHRF1 depletion to sensitize KRAS mutant cancer cells towards KRAS G12C inhibitors or MEK kinase inhibitors.

Minor points

1. Figure 1: Does the mouse cell line LKR10 used in 2D screen also have Tp53 loss?

2. Figure 2b: Indel frequency seems to indicate that in most cell lines UHRF1 knockout was heterozygous. However, in Figure 2a UHRF1 protein level seems to be reduced by more than 50%. I wonder why this is the case. Also, UHRF1 CRISPR KO WB for H1437 and NL20 cell lines should be shown.

3. Figure S3e: UHRF1 knockdown also caused expansion of S phase. Did the UHRF1 KD cells have higher levels of DNA damage?

4. Figure 4: Among the 80 TSGs that became hypo-methylated and up-regulated in UHRF1 depleted cells, are there any TSGs that have been previously shown to be hyper-methylated and repressed in a KRAS-dependent manner? This might further narrow down the candidates that are responsible for the growth inhibition in UHRF1 knockdown cells.

5. Figure S3a right panel: cell line label's color did not match. The first 4 cell lines' label should be red?

6. Figure S2d: a breakdown of the labels by cell line names would be helpful.

7. Figure 4d and S4e: what are the genes highlighted in green?

Reviewer #2 (Remarks to the Author):

General Comments:

This manuscript contains potentially important data for ascribing a key role for UHRF1, a critical protein for control of DNA methylation in cancer, and relationship of the findings to KRAS and p53 mutations in non-small cell lung cancer (NSCLC) of the lung adenocarcinoma (LUAD) type. The approach of agnostically coming to UHRF1 as the above important gene by performing a synthetic lethal screen approach seems well formulated and performed and the paper is clearly written. This said, I have some major questions about the analyses utilized and some conclusions reached which I believe mandate some additional examination of these to allow the biology of the findings to be fully evaluable and which are of most significance to the cancer research community. My concerns relate to dissecting further the mechanisms to explain the interesting phenotype obtained by the authors. My specific suggestions and queries are as follows:

Specific suggestions and Comments:

1. Mechanisms involved: the authors reach a reasonable conclusion that the major mechanistic explanation for the effects of UHRF1 loss in the systems examined could result from involve losses of DNA methylation that result in upregulated expression of tumor suppressor genes (TSG's) which have been abnormally silenced in association with promoter DNA methylation. However, this mechanism is certainly not fully clarified in the work as it stands and especially for the associations of genetic changes integral to the work. Also, as introduced earlier above, while certainly, the findings have much potential links to mutations of KRAS and p53 mutations with respect to TSG's, it is enigmatic that the authors do not point to the relevance of the work for immune checkpoint therapy (IO). Occurrence of KRAS and when co-mutated with p53 has very important implications for IO and specifically for LUAD – these are being discussed in the literature, and associated with

epigenetic mechanisms and analogies to viral defense changes, and this should be highlighted and commented upon. The authors may need to perform several more genome-wide DNA methylation and RNA-seq experiments in both KRAS mutant and WT cells to dissect the differential requirement of UHRF1. Again, more on this below.

2. TSG's identified and correlations presented for signaling pathway consequences: It is enigmatic when one goes over the list of reactivated TSG's reported by the authors, most of the most driver events well reported for LUAD are not found such as for p16 etc? The authors are urged to look at the first TCGA paper for LUAD and clarify this. Further, the authors state that there is little overlap with TSG's silenced epigenetically in colorectal cancer (CRC) versus LUAD so UHRF1 may work differently for these two malignancies. . This seems strange as many TSG's overlap between cancers like LUAD and CRC even though, certainly there can be cancer specific genes. Furthermore, this will all affect pathway analyses which also seem enigmatic in the author's data. It is well reported by several groups that some 60 to 80% of genes with abnormal, promoter, CpG island methylation in cancer are genes mediated by polycomb group proteins (PCG) in development but without promoter DNA methylation. Usually, then this is reflected in the pathways associated with these genes as reflected by appearance of developmental pathways, cancer pathways, stem cell relationships etc. With respect specifically to LUAD, again the authors are urged to look at the TCGA paper mentioned above and others. Finally, relative to the relationships of immune changes with respect to DNA methylation changes, these generally can be seen with induced demethylation in multiple cancer types and such pathways generally show up as prominent because of this. Again, this is highly relevant for importance for IO therapies as mentioned earlier above.

Reviewer #3 (Remarks to the Author):

In this manuscript entitled "UHRF1 is a mediator of KRAS driven oncogenesis in lung adenocarcinoma", Kostyrko and colleagues identify that UHRF1 is downstream of KRAS and a novel candidate targeting KRAS mutant lung cancer according to their shRNA screening utilizing 3D spheroid culture of primary mouse lung cancer cells. Consistent with the function of UHRF1 as an epigenetic regulator involved in DNA methylation, they observe UHRF1 depletion results in global DNA hypomethylation leading to up-regulation of tumor suppressor genes which regulate AKT/mTOR, WNT/ β -catenin pathway etc., and subsequent growth arrest and apoptosis in KRAS mutant lung cancer. Interestingly, differentially expressed genes following UHRF1 depletion and KRAS depletion are overlapped to some degree, suggesting UHRF1 is a part of the downstream pathway of oncogenic KRAS. They also find that high UHRF1 expression correlates with poor prognosis in KRAS mutant lung adenocarcinoma patients.

The manuscript is overall well written and the study sheds light on the novel KRAS function using both in vitro and in vivo models while providing avenues for therapeutic interventions. Although they mentioned it in the discussion, the main cavity of this paper is the mechanisms by which oncogenic KRAS especially regulates UHRF1 function among other epigenetic regulators, I think several points should be addressed to strengthen their findings before its publication in a journal such as Nature Communications. I have listed below some constructive suggestions:

1. The molecular mechanisms by which oncogenic KRAS regulates UHRF1 are not well-described. Given that the methylation changes following KRAS activation is independent of the canonical pathway such as MAPK as the authors expected, do treatment of KRAS inhibitors such as sotorasib or adagrasib particularly impact UHRF1 expression and tumor suppressor proteins associated with hypomethylation, compared with MEK inhibitors or PI3K inhibitors? I think it may be a valuable approach to investigate the differential clinical response between KRAS G12C inhibitors and for example MEK inhibitors in the patients. Since they utilized H358 cells, which are sensitive cells to sotorasib treatment, they should strengthen their models by using clinically relevant compounds in addition to genetic ablation.

2. Recently, many papers have reported that treatment of epigenetic inhibitors including DNMT inhibitor dysregulates histone/DNA modification of noncoding regions including endogenous retrovirus (ERVs), contributing to tumor suppressive function (e.g. PMID:26317465, 26317466). Given that UHRF1 depletion induces global hypomethylation of DNA in KRAS mutant lung cancer,

the authors could investigate the methylation status around ERVs and/or other noncoding regions in addition to the expression of tumor suppressor genes. Indeed, in Extended figure 4, it appears that the JAK/STAT pathway is also activated following UHRF1 depletion, which is one of the downstream pathways of ERVs activation.

3. The authors state that UHRF1 is KRAS downstream, and KRAS mutant lung cancer cell lines highly express UHRF1 compared with KRAS wild-type lung cancer cell lines by using several cell lines. Is there any evidence to reveal the correlation between oncogenic KRAS and UHRF1 expression in lung cancer patients? They show TCGA data in figure 6C (right), and they should mention statistical significance.

4. Related with the above discussion, does introduction of oncogenic KRAS in KRAS wild-type lung cancer cells up-regulate UHRF1 expression?

Reviewer 1

Major points:

1. For the primary RNAi screen, the authors compared Kras:Tp53 mutant primary lung tumor cells cultured in Matrigel-based 3D culture vs. a Kras mutant (but not Tp53 mutant?) cell line cultured in standard media in 2D culture. The use of different cell lines and different culture media under these two conditions could complicate the result by introducing additional variables. As collateral dependency genes are often context dependent, their essentiality might be influenced by media and genetic background of the cells.

As the reviewer correctly points out, the cells used for 3D culture and the ones used in 2D are not identical. Ideally, we would have grown the primary mouse tumor cells in both conditions. However, primary tumor cells from the KP model do not grow when removed from a mouse and plated in standard 2D culture conditions. We therefore chose as a comparison (rather than as a control), the well-known LKR10 cells which were derived from a similar KRAS-mutated murine lung tumor model. We reasoned that a comparison to a cell line grown in 2D culture would identify alterations found in both 2D and 3D and nominate those more likely to be 3D-specific (but not ruling out that these were cell-type specific and not culture condition specific). With regards to the p53 status of LKR10 cells, it is correct that these are Trp53 wild-type (see immunoblot below).

Despite this caveat, we respectfully suggest that the extensive subsequent validation of these initial results in a panel of KRAS mutant human lung cancer cell lines, some of which are TP53 wild-type (A549, H358, H460) and some of which are TP53 mutant (H2009, H23, H1792) and which of course have different genetic backgrounds revealed that the observed effect is not p53 dependent or context specific but rather is found in a variety of genetic backgrounds in both mouse and human. Therefore, while there were some unavoidable limitations to the original screen, the extensive validation in different cell lines and across species mitigates the potential context-dependence which is a well-known limitation of essentiality screens.

Figure 1 LKR10 cells treated with 0, 0.5 or 1uM of doxorubicin. Trp53 wild-type mouse cells (imr90) used as control.

2. UHRF1 is likely to be an essential gene in normal cells due to its role in CpG methylation and cell cycle progression. The authors showed that partial loss of UHRF1 function, through either siRNA knockdown or incomplete CRISPR KO, causes greater dependencies in 3D culture. Why cancer cells in spheroid culture are more sensitive to the perturbation of UHRF1 than those in 2D culture was not well articulated. If I understood the methods correctly, gene expression and gene

methylation studies were conducted using cells in 2D culture, and it is unclear whether these changes are also observed in 3D culture.

We thank the reviewer for pointing out the need for further clarification here. The results of our shRNA screens performed in 3D and 2D as well as the RNAi screen data from DepMap and Project DRIVE performed in 2D do suggest that knock-down of UHRF1 is more deleterious in 3D than in 2D. However, we subsequently found that a complete knock-out of UHRF1, either via CRISPR (which leads to a complete loss of the UHRF1 protein as evidenced by western blots in Figure 2a) or Cre recombinase in the mouse model has profound anti-proliferative effects in KRAS mutant lung cancer cells in vivo and in vitro both in 3D and in 2D cultures (as shown in Extended Data Figure 2b-d and in Extended Data Figure 6g). Therefore, we reasoned that gene expression and methylation changes observed in 2D cultures would be relevant to the mechanisms driving the observed essentiality of UHRF1 loss in the context of KRAS. This was the rationale for carrying out the additional RNAseq and methylation studies in 2D. Importantly, cell lines derived from normal human bronchial epithelium (NL20 and BEAS2B) showed significantly less dependence on UHRF1 than did KRAS-mutant cancer cells, indicating that UHRF1 is not essential for normal cells (see Fig 2b and Extended Data Fig. 2c-e).

3. The authors presented evidence that UHRF1 knockdown is more deleterious to KRAS mutant cells compared to KRAS WT cells. Do the KRAS WT cells used in this study harbor EGFR mutation? Mechanistically, it was not clear as to why UHRF1 depletion is particularly toxic in KRAS mutant cells. Consistent with previous reports, a subset of ~80 TSGs were up-regulated in UHRF1-deleted cells. It would be useful to test whether one or more such TSGs are responsible for the phenotype, for example, by carrying out double KO experiments.

We thank the reviewers for the helpful suggestion that we should further evaluate which TSGs are most relevant to the observed phenotype of UHRF1 loss in KRAS mutant cells. Rather than evaluate this using a candidate approach, we opted to carry out a small focused CRISPR screen. These extensive additional experiments are now included in the manuscript (Figure 5 and Extended Data Figure 5) and described in the text. We reasoned that sgRNAs targeting TSGs downregulated by UHRF1 would show positive selection after UHRF1 knockdown but would be under neutral selection in cells infected with a control sgRNA. As shown in Figure 5, we identified several TSGs that have functional importance after UHRF1 knockdown using this approach. These additional studies extend the functional relevance of our findings and nominate key TSGs for further evaluation in LUAD.

With regards to the question of EGFR mutations, two KRAS wt cell lines used in this study harbor an EGFR mutation (H1975 and H1650) and these cell lines were excluded from most experiments. The remaining KRAS wt cell lines do not carry EGFR mutations.

4. The authors showed that UHRF1 KO cells had higher levels of apoptosis and elevated Trail expression. Trail signaling is known to induce apoptosis. It would be useful to investigate whether cell death is caused by Trail up-regulation or a Trail-independent mechanism. Also, do the UHRF1 KO cells exhibit higher levels of DNA

damage in KRAS mutant cells compared to KRAS WT cells?

To address the reviewer's question regarding the potential effect of UHRF1 KO on DNA damage, we evaluated the levels of the DNA damage marker γ H2AX (pH2AX), in two KRAS mutant cell lines treated with a control or anti-UHRF1 siRNA. Quantification of pH2AX foci in IF images demonstrated an approx. 2-fold increase in the number of foci per cell in cells treated with the UHRF1 siRNA. We hypothesize that this effect may be due to the role of UHRF1 in regulating homologous recombination (as described by Zhang, Liu et al., Nat Comms, 2016), which likely leads to delayed progression through S phase, in line with our cell cycle analysis (Extended Data Fig. 3c-e). These new experiments have now been included in Extended Data Fig. 3g-j.

5. Genetically it has been shown that, with regard to DNA methylation, UHRF1 loss can phenocopy DNMT1 loss. To further explore the translational potential of this work, the authors could attempt to target DNMT1 using inhibitors (for example, GSK3685032) and evaluate whether DNMT1 inhibition is effective at attenuating the growth of KRAS mutant lung cancer cells in vitro and in vivo. In addition, if the primary mechanism of UHRF1 depletion is TSG upregulation, it might be expected that UHRF1 depletion to sensitize KRAS mutant cancer cells towards KRAS G12C inhibitors or MEK kinase inhibitors.

We thank the reviewer for these excellent suggestions. To address the first point we treated two KRAS mutant lung cancer cell lines (H358 and A549) cultured in 3D with the DNMT1 inhibitor GSK3685032. In both cell lines the inhibitor elicited an effect very similar to that of UHRF1 knock-out. This result has now been included in Figure 3d.

To address the second point we treated control or UHRF1 depleted cells with the MEK inhibitor trametinib (H358 and A549 cells) or a KRASG12C inhibitor Sotorasib (H358 cells). In addition we also treated both cell lines with a PI3K inhibitor Copanlisib. The results (included in a Figure 3c-d and Extended Data Figure 2f) demonstrate that UHRF1 loss does sensitize KRAS mutant cells to either direct KRAS inhibition, MEK inhibition or PI3K inhibition.

Minor points

1. Figure 1: Does the mouse cell line LKR10 used in 2D screen also have Tp53 loss? LKR10 do not have Trp53 loss (see response to Major point 1 above).

2. Figure 2b: Indel frequency seems to indicate that in most cell lines UHRF1 knockout was heterozygous. However, in Figure 2a UHRF1 protein level seems to be reduced by more than 50%. I wonder why this is the case. Also, UHRF1 CRISPR KO WB for H1437 and NL20 cell lines should be shown.

Western blots for UHRF1 protein in all cell lines, including H1437 and NL20, have now been added to Figure 2a. Results of the TIDE analysis (previously shown in Figure 2b) have been removed, as appear to have been somewhat misleading. It should be noted that TIDE is an in-silico estimation of CRISPR knock-out efficiency, limited to short in-dels and dependent on the quality of Sanger sequencing, thus it may inaccurately predict the presence/absence of the targeted protein.

3. Figure S3e: UHRF1 knockdown also caused expansion of S phase. Did the

UHRF1 KD cells have higher levels of DNA damage?

To answer this we performed IF for pH2AX in two KRAS mutant cell lines treated with control or UHRF1 siRNA (data now included in Extended Data Fig. 3g-j), which revealed an approx. 2-fold increase in pH2AX foci per cell in both cell lines. This increase is consistent with the extended S phase in the UHRF1-depleted cells.

4. Figure 4: Among the 80 TSGs that became hypo-methylated and up-regulated in UHRF1 depleted cells, are there any TSGs that have been previously shown to be hyper-methylated and repressed in a KRAS-dependent manner? This might further narrow down the candidates that are responsible for the growth inhibition in UHRF1 knockdown cells.

Following the reviewer's suggestion, we compared the list of genes hypomethylated in our UHRF1-depleted cells with the list of 50 genes hypermethylated in lung cells expressing mutant KRAS reported in a recent study by Tew et al (Sci Reports, 2022), and found only one gene (HNF1B) in common. This low level of overlap is not particularly surprising, as the authors also noted a very limited overlap between the cell lines they tested. While we did not identify individual genes in common, our pathway analysis on differentially methylated CpGs showed a significant enrichment in pathways involved in development, morphogenesis and differentiation, consistent with the authors' observations from their dataset.

Finally, as noted in point #3 above, we have now carried out a focused CRISPR screen of the 80 identified TSGs in order to narrow down the candidates in an unbiased manner.

5. Figure S3a right panel: cell line label's color did not match. The first 4 cell lines' label should be red?

This has now been corrected.

6. Figure S2d: a breakdown of the labels by cell line names would be helpful.

A legend with cell line names has now been added to the plot in Extended data Figure 2d.

7. Figure 4d and S4e: what are the genes highlighted in green?

Genes highlighted in green are examples of significantly over-expressed TSGs. This has now been clarified in the legends and captions to these figures. In addition UHRF1 and KRAS have now been indicated in blue (previously also in green).

Reviewer 2

1. Mechanisms involved: the authors reach a reasonable conclusion that the major mechanistic explanation for the effects of UHRF1 loss in the systems examined could result from involve losses of DNA methylation that result in upregulated expression of tumor suppressor genes (TSG's) which have been abnormally silenced in association with promoter DNA methylation. However, this mechanism is certainly not fully clarified in the work as it stands and especially for the associations of genetic changes integral to the work. Also, as introduced earlier above , while certainly, the findings have much potential links to mutations of KRAS and p53 mutations with respect to TSG's, it is enigmatic that the authors do not point to the relevance of the work for immune checkpoint therapy (IO). Occurrence of KRAS and when co-mutated with p53 has very important implications for IO and specifically for LUAD – these are being discussed in the literature, and associated with epigenetic mechanisms and analogies to viral defense changes, and this should be highlighted and commented upon. The authors may need to perform several more genome-wide DNA methylation and RNA-seq experiments in both KRAS mutant and WT cells to dissect the differential requirement of UHRF1. Again, more on this below.

We thank the reviewer for highlighting the needs for additional functional studies to clarify the role of TSGs upregulated after UHRF1 knock-out. As noted in point #3 above, we have now carried out extensive additional studies using a pooled sgRNA screen to identify critical TSGs regulated by UHRF1. These studies are now described in Figure 5 and Extended Data Figure 5. With regards to the question of IO therapy, this is an excellent point from the reviewer and an omission on our part. The relevance of the present study to immune checkpoint therapy has now been added into the discussion.

2. TSG's identified and correlations presented for signaling pathway consequences: It is enigmatic when one goes over the list of reactivated TSG's reported by the authors, most of the most driver events well reported for LUAD are not found such as for p16 etc? The authors are urged to look at the first TCGA paper for LUAD and clarify this.

We thank the reviewer for this interesting suggestion. To evaluate the overlap between genes upregulated by UHRF1 and those previously noted as LUAD TSGs, we downloaded the original LUAD publication (PMC4231481) which contained 70 genes. From this we subsetted those annotated as TSG in the COSMIC database, which gave a total of 16 genes. Interestingly, of these 12 were not in our original TSG list, including the p16 gene (CDKN2A). The newly added genes are: ARID1A, ARID2, ATM, CCDC6, CDKN2A, CLTC, KEAP1, NF1, RBM10, SLC34A2, SMARCA4, TP53. None of the genes in this list were correlated with UHRF1 expression in the cell lines analyzed here. However, it should be noted that many of these genes are known to be inactivated in lung cancer via other mechanisms including loss-of-function mutation or deletion. In the cell lines used here H358 cells have TP53 loss, KEAP1 loss, SMARCA4 loss, CDKN2A loss and A549 cells have KEAP1 loss, ARID1A loss, ATM loss, CDKN2A loss, and SMARCA4 loss. This may explain why these genes were not identified among the genes reactivated

by UHRF1 depletion. As noted in point #3, we have now validated a number of the UHRF1 upregulated genes as potential LUAD TSGs.

Further, the authors state that there is little overlap with TSG's silenced epigenetically in colorectal cancer (CRC) versus LUAD so UHRF1 may work differently for these two malignancies. This seems strange as many TSG's overlap between cancers like LUAD and CRC even though, certainly there can be cancer specific genes. Furthermore, this will all affect pathway analyses which also seem enigmatic in the author's data. It is well reported by several groups that some 60 to 80% of genes with abnormal, promoter, CpG island methylation in cancer are genes mediated by polycomb group proteins (PCG) in development but without promoter DNA methylation. Usually, then this is reflected in the pathways associated with these genes as reflected by appearance of developmental pathways, cancer pathways, stem cell relationships etc. With respect specifically to LUAD, again the authors are urged to look at the TCGA paper mentioned above and others. Finally, relative to the relationships of immune changes with respect to DNA methylation changes, these generally can be seen with induced demethylation in multiple cancer types and such pathways generally show up as prominent because of this. Again, this is highly relevant for importance for IO therapies as mentioned earlier above.

In the original manuscript we only show a subset of the highest ranking pathways and we agree that many of the results were obscured as a result. To clarify this point we now included a supplemental excel sheet (Supplementary table 5) that with the full pathway analysis and a figure showing pathways grouped by similarity (Figure 4b). As shown in the figure, many of the aforementioned pathways are indeed represented including those involved in stem cells, development, and morphogenesis.

Reviewer 3

1. The molecular mechanisms by which oncogenic KRAS regulates UHRF1 are not well-described. Given that the methylation changes following KRAS activation is independent of the canonical pathway such as MAPK as the authors expected, do treatment of KRAS inhibitors such as sotorasib or adagrasib particularly impact UHRF1 expression and tumor suppressor proteins associated with hypomethylation, compared with MEK inhibitors or PI3K inhibitors? I think it may be a valuable approach to investigate the differential clinical response between KRAS G12C inhibitors and for example MEK inhibitors in the patients. Since they utilized H358 cells, which are sensitive cells to sotorasib treatment, they should strengthen their models by using clinically relevant compounds in addition to genetic ablation.

We thank the reviewer for these helpful suggestions. Following this recommendation, we treated the KRASG12C cell line H358 with MEKi trametinib, KRAS G12Ci sotorasib, or PI3Ki copanlisib and blotted for UHRF1 protein. We found that treatment with both trametinib and sotorasib leads to a significant decrease of UHRF1 protein, while treatment with copanlisib did not significantly impact UHRF1 protein levels. In sotorasib- and trametinib-treated cells we also detected an almost complete loss of Cyclin D1 (CCND1), suggesting that these two treatments lead to cell cycle arrest in G1 phase (as already shown in Extended Data Fig. 3c-e), which may indicate that at least part of the effect is the result of cell cycle arrest. This data has now been included in Extended data figure 3f.

2. Recently, many papers have reported that treatment of epigenetic inhibitors including DNMT inhibitor dysregulates histone/DNA modification of noncoding regions including endogenous retrovirus (ERVs), contributing to tumor suppressive function (e.g. PMID:26317465, 26317466). Given that UHRF1 depletion induces global hypomethylation of DNA in KRAS mutant lung cancer, the authors could investigate the methylation status around ERVs and/or other noncoding regions in addition to the expression of tumor suppressor genes. Indeed, in Extended figure 4, it appears that the JAK/STAT pathway is also activated following UHRF1 depletion, which is one of the downstream pathways of ERVs activation.

To address this point we downloaded and selected for ERV regions from UCSC RepeatMasker. We then intersected all EPIC probes that were within these regions and repeated the analysis similar to that of shown on Extended data figure 4b. The new figure (see below) shows a similar global decrease in ERV-specific methylation only in the siUHRF1 group and not siKRAS group. Although this data suggests that ERV elements may be affected as well, the overall decrease is most likely a global genomic event brought about by the siUHRF1 perturbation.

Figure 2 Methylation of ERV-containing probes present in the EPIC array dataset.

3. The authors state that UHRF1 is KRAS downstream, and KRAS mutant lung cancer cell lines highly express UHRF1 compared with KRAS wild-type lung cancer cell lines by using several cell lines. Is there any evidence to reveal the correlation between oncogenic KRAS and UHRF1 expression in lung cancer patients? They show TCGA data in figure 6C (right), and they should mention statistical significance.

We added a p-value ($p=0.69$) to figure 6c (now: Figure 8a) comparing UHRF1 expression between KRAS mutant vs KRAS wt LUAD patients. While there is no statistical difference between these groups, we do note that UHRF1 expression is significantly correlated with KRAS expression in patient samples. When we stratify the LUAD dataset into KRAS mut and KRAS wt, UHRF1 expression significantly correlates with KRAS expression in both groups, with a slightly bigger coefficient in the KRAS mut group (KRASmut: $r=0.37$, $p=0.0000039$ vs KRASwt: $r=0.21$, $p=0.000084$). These plots have now been included in Extended Data Fig. 3b.

In addition, indirect evidence for the correlation between UHRF1 and KRAS is the finding that 323 out of 450 TSGs (72%) significantly anti-correlated with UHRF1 expression in patient samples also has a significant correlation with KRAS, with the vast majority of them (291/323 or 90%) being anti-correlated with KRAS expression. Of note, our survival analysis shows that 23 of these genes has a significant impact on overall survival only in KRAS mut group and not KRAS wt group (see: Extended Data Figure 7b).

4. Related with the above discussion, does introduction of oncogenic KRAS in KRAS wild-type lung cancer cells up-regulate UHRF1 expression?

To answer this question, we transfected three KRAS wt cell lines (H1437, NL20, and BEAS2B) with a plasmid expressing a GFP-tagged KRAS G12C protein. Western blot of cell extracts 72h post transfection revealed no changes in UHRF1 protein levels. This suggests that, at least within the timeframe of this experiment, KRAS overexpression does not lead to upregulation of UHRF1. This result had now been included in Extended Data Figure 7d and elaborated upon in the discussion.

REVIEWER COMMENTS

Reviewer #1 (Remarks to the Author):

In this revision, Kostyrko and colleagues presented new experiments and additional evidence to support the synthetic lethal role of the epigenetic regulator UHRF1 in KRAS mutant lung cancer. The authors showed that UHRF1 knockdown induced more DNA damage and resulted in the upregulation of a subset of tumor suppressor genes (TSGs). The authors further clarified the mechanism by showing that down-regulation of a small number of UHRF1-dependent TSGs can rescue the viability defect in UHRF1 knockdown cells. The authors also showed that inhibitor of the DNA methyltransferase DNMT1, a binding partner of UHRF1, can phenocopy UHRF1 knockdown and cooperate with KRAS inhibitor to reduce the viability of KRAS mutant cells in 3D spheroid culture. The revision has substantially improved the study and the authors have addressed the key concerns I raised. I recommend the study to be accepted for publication pending minor revision outlined below.

Minor points

1. Figure 3d, right panel, label incorrect for the pink bar, should be sgNeg+DNMT1i?
2. Figure 6g, the % of EdU+Uhrf1 double positive cells in the UKP lung tissue represents those that escaped Uhrf1 deletion? It wasn't clear which cells were being measured here.

Reviewer #2 (Remarks to the Author):

The authors certainly have addressed some of our concerns, especially regarding the mechanism part and performed considerable new bench and informatics work. Specifically, they performed a focused CRISPR screening and identified 15 DNA hypermethylated TSGs whose knockout rescued the impaired growth induced by UHRF1 KD in KRAS mutant cancer (Fig 5 of the revised manuscript). They further showed some of the key genes were negatively correlated with UHRF1 in TCGA LUAD analysis. Nevertheless, p16 was not included in their TSG list, which might be due to the somatic inactivation of p16 in their models. The authors did the GSEA analysis showing the enrichment of developmental pathway/genes by using the DNA methylation data, and discussed the relationship between UHRF1, TSG silencing (e.g. KEAP1), and immune checkpoint therapy (IO) in KRAS mutant lung cancer.

However, several of the new findings and responses said to address concerns are a bit puzzling. First although they list upregulated genes with UHRF1 knock out, no specific DNA methylation is defined for the individual genes. Moreover, as pointed out in the reviews, many there are many well know hypermethylated genes in NSCLC, that also overlap with those in colon cancer. They list genes other than these many of which don't fit these above categories. Seems most doable to show the methylation of the promoters of these genes and upregulation by specific PCR. Similarly, the authors were suggested to comment on the reactivation of viral defense pathways after UHRF1 inhibition to potentiate the IO in KRAS mutant lung cancer and were sked about ERV's. No viral defense genes are mentioned and while they give an answer about ERV's, this is again puzzling as to whether they looked at whole families of these repeats or at some specific ERV's as they state "Although this data suggests that ERV elements may be affected as well, the overall decrease is most likely a global genomic event brought about by the siUHRF1 perturbation". Also, the authors didn't perform additional genome-wide DNA methylation and RNA-seq data in KRAS WT lung cancer cells as suggested. The data in the revised manuscript still didn't dissect the differential requirement of UHRF1 in KRAS mutant and WT cells. Therefore, I think the authors might still consider the additional work suggested.

Reviewer #3 (Remarks to the Author):

Although some questions remain, the authors have submitted a revised manuscript that includes a number of new experiments and has been strengthened. The authors have satisfactorily addressed the main points raised by the reviewers and the work is acceptable for publication.

Response to reviewer critiques:

We once again thank the reviewers for their careful reading of our revised manuscript and we are gratified to see that we have satisfactorily addressed the majority of the prior critiques. Below we detail how the remaining critiques have been addressed. In most cases, this required only revisions to the text or figures. Where experiments were requested, we have performed some additional experiments or explained why in our opinion they are outside the scope of the current work.

Reviewer #1:

In this revision, Kostyrko and colleagues presented new experiments and additional evidence to support the synthetic lethal role of the epigenetic regulator UHRF1 in KRAS mutant lung cancer. The authors showed that UHRF1 knockdown induced more DNA damage and resulted in the upregulation of a subset of tumor suppressor genes (TSGs). The authors further clarified the mechanism by showing that down-regulation of a small number of UHRF1-dependent TSGs can rescue the viability defect in UHRF1 knockdown cells. The authors also showed that inhibitor of the DNA methyltransferase DNMT1, a binding partner of UHRF1, can phenocopy UHRF1 knockdown and cooperate with KRAS inhibitor to reduce the viability of KRAS mutant cells in 3D spheroid culture. The revision has substantially improved the study and the authors have addressed the key concerns I raised. I recommend the study to be accepted for publication pending minor revision outlined below.

We thank the reviewer for their prior suggestions and we are gratified that they recommend that this work be published in *Nature Communications* pending the minor revisions we have now completed as described below.

We address the following remaining minor points:

1. Figure 3d, right panel, label incorrect for the pink bar, should be sgNeg+DNMT1i? Indeed, a “+” was missing under DNMT1i treatment. This has now been corrected in the figure.

2. Figure 6g, the % of EdU+Uhrf1 double positive cells in the UKP lung tissue represents those that escaped Uhrf1 deletion? It wasn't clear which cells were being measured here.

The figure represents all double positive cells detected in UKP mouse lungs. Given that we did not identify any tumor lesions that were Uhrf1-negative in UK and UKP mice post AdCre treatment (as mentioned in Results section lines 265-267), we hypothesize that all EdU+Uhrf1 double positive cells shown in Figure 6g represent those that escaped Uhrf1 deletion. Additional clarification has been added to the text related to Figure 6g (lines 279-282).

Reviewer #2:

The authors certainly have addressed some of our concerns, especially regarding the mechanism part and performed considerable new bench and informatics work. Specifically, they performed a focused CRISPR screening and identified 15 DNA hypermethylated TSGs whose knockout rescued the impaired growth induced by UHRF1 KD in KRAS mutant cancer (Fig 5 of the revised manuscript). They further showed some of the key genes were negatively correlated with UHRF1 in TCGA LUAD analysis. Nevertheless, p16 was not included in their TSG list, which might be due to the somatic inactivation of p16 in their models. The authors did the GSEA analysis showing the enrichment of developmental pathway/genes by using the DNA methylation data, and discussed the relationship between UHRF1, TSG silencing (e.g. KEAP1), and immune checkpoint therapy (IO) in KRAS mutant lung cancer.

However, several of the new findings and responses said to address concerns are a bit puzzling. First although they list upregulated genes with UHRF1 knock out, no specific DNA methylation is defined for the individual genes. Moreover, as pointed out in the reviews, many there are many well know hypermethylated genes in NSCLC, that also overlap with those in colon cancer. They list genes other than these many of which don't fit these above categories. Seems most doable to show the methylation of the promoters of these genes and upregulation by specific PCR.

We again thank the reviewer for carefully reading our manuscript. We respectfully suggest that in most cases, validation of RNAseq with PCR is considered to be unnecessary as the former is highly quantitative and in addition does not suffer from some of the limitations of PCR, such as low-throughput or dependance on primer quality. However, as requested, we performed PCR validation of some of the UHRF1-specific hits identified in the CRISPR screen using two KRAS mutant cell lines (H358, A549) and two KRAS wild type cell lines (H1437, NL20) for comparison. These data are now added to Figure 5e. In addition, we also show the gene expression of all 15 UHRF1-specific hits (derived from RNA sequencing of H358 and A549 cells) in Extended data figure 5c. Moreover, we also show the hypomethylated probes in the promoter regions of the 15 genes identified in the CRISPR screen on Figure 5f and Extended data figure 5d. We are confident that this satisfactorily addresses the remaining concerns of the reviewer with regards to this point.

With regards to the overlap between lung cancer and colorectal cancer TSGs, we find, as mentioned in Discussion (lines 365-367) little overlap between TSGs upregulated with UHRF1 loss in our lung cancer models compared to those identified in colorectal cancer cell lines by Kong and colleagues (Kong et al, *Cancer Cell*, 2019) – see venn diagram below. While outside the scope of our work, we hypothesize that this is related to other, likely tissue specific, layers of epigenetic regulation of which UHRF1 is only one aspect.

Figure 1 Overlapp between TSGs hypomethylated with UHRF1 loss in lung cancer cells (LUAD) in present study and those hypomethylated and upregulated in Kong et al (Table S1 Kong et al, Cancer Cell, 2019).

Similarly, the authors were suggested to comment on the reactivation of viral defense pathways after UHRF1 inhibition to potentiate the IO in KRAS mutant lung cancer and were sked about ERV's. No viral defense genes are mentioned and while they give an answer about ERV's, this is again puzzling as to whether they looked at whole families of these repeats or at some specific ERV's as they state "Although this data suggests that ERV elements may be affected as well, the overall decrease is most likely a global genomic event brought about by the siUHRF1 perturbation".

In our previous attempt to address the Reviewer's question we utilized all available endogenous retrovirus (ERV) coordinates from the UCSC RepeatMasker database without applying any filters or selective processes. While the Reviewer's suggestion is very intriguing and certainly merits further investigation, we refrained from discussing it further in our work for two main reasons: 1) because it falls outside the scope of our study (which is focused on protein coding genes) and 2) because we lack the necessary expertise to draw a detailed conclusion regarding specific ERVs. As a result, we opted to only display the general trend. We respectfully suggest that additional evaluation of ERVs does not influence the conclusions of our manuscript and therefore is most appropriate for a separate study.

Also, the authors didn't perform additional genome-wide DNA methylation and RNA-seq data in KRAS WT lung cancer cells as suggested. The data in the revised manuscript still didn't dissect the differential requirement of UHRF1 in KRAS mutant and WT cells. Therefore, I think the authors might still consider the additional work suggested.

In carefully considering this suggestion, we believe we somewhat overstated our conclusions leading to this comment, for which we apologize. What we show in our work is that UHRF1 knock-down in mouse and human models is a specific vulnerability in KRAS mutant NSCLC as shown through multiple assays. We then perform RNAseq analysis, global methylation analysis and a focused CRISPR screen on KRAS mutant

cells in an effort to mechanistically dissect this KRAS-specific effect. Our conclusion is that UHRF1 represses a set of tumor suppressor genes which together are important for KRAS-driven oncogenesis (as validated by the CRISPR screen). We anticipate that UHRF1 knock-down also has some effects on transcription and methylation in KRAS wildtype cells, but since we clearly show that functionally this does not have a phenotype in these cells, these UHRF1-dependent TSGs are specifically important in the KRAS mutant context. Therefore, we respectfully suggest that the additional extensive work suggested (methylation and RNA-seq on KRAS wild-type cells) is outside the scope of our work and not relevant to the conclusions. We have carefully revised several statements throughout the manuscript to narrow the scope of our conclusions and make this point clear and we thank the reviewer for helping us identify the need to clarify our claims.

Reviewer #3:

Although some questions remain, the authors have submitted a revised manuscript that includes a number of new experiments and has been strengthened. The authors have satisfactorily addressed the main points raised by the reviewers and the work is acceptable for publication.

We thank the reviewer for their prior suggestions and we are gratified that they now find this work acceptable for publication in *Nature Communications*.